# PEG-stabilized coaxial stacking of two-dimensional covalent organic frameworks for enhanced photocatalytic hydrogen evolution

Ting Zhou[1,6], Lei Wang[2,6], Xingye Huang[1], Junjuda Unruangsri[3], Hualei Zhang[1], Rong Wang[1], Qingliang Song[1], Qingyuan Yang [4], Weihua Li[1], Changchun Wang[1], Kaito Takahashi [5✉], Hangxun Xu [2✉] & Jia Guo[1✉]

Two-dimensional covalent organic frameworks (2D COFs) featuring periodic frameworks, extended π-conjugation and layered stacking structures, have emerged as a promising class of materials for photocatalytic hydrogen evolution. Nevertheless, the layer-by-layer assembly in 2D COFs is not stable during the photocatalytic cycling in water, causing disordered stacking and declined activity. Here, we report an innovative strategy to stabilize the ordered arrangement of layered structures in 2D COFs for hydrogen evolution. Polyethylene glycol is filled up in the mesopore channels of a β-ketoenamine-linked COF containing benzothiadiazole moiety. This unique feature suppresses the dislocation of neighbouring layers and retains the columnar π-orbital arrays to facilitate free charge transport. The hydrogen evolution rate is therefore remarkably promoted under visible irradiation compared with that of the pristine COF. This study provides a general post-functionalization strategy for 2D COFs to enhance photocatalytic performances.

[1] State Key Laboratory of Molecular Engineering of Polymers, Department of Macromolecular Science, Fudan University, Shanghai, China. [2] Hefei National Laboratory for Physical Sciences at the Microscale, CAS Key Laboratory of Soft Matter Chemistry, University of Science and Technology of China, Hefei, China. [3] Department of Chemistry, Chulalongkorn University, Bangkok, Thailand. [4] State Key Laboratory of Organic-Inorganic Composites, Beijing University of Chemical Technology, Beijing, China. [5] Institute of Atomic and Molecular Sciences Academia Sinica, Taipei, Taiwan. [6]These authors contributed equally: Ting Zhou, Lei Wang. ✉email: kt@gate.sinica.edu.tw; hxu@ustc.edu.cn; guojia@fudan.edu.cn

Conjugated polymers are a promising class of semi-conductors for converting solar energy into chemical fuels such as photocatalytic production of hydrogen[1,2]. The electronic structures of polymeric photocatalysts could be designed to increase absorption in the visible window to adjust bandgaps and orbital levels, as well as to improve charge separation and transport. Hence, the hydrogen evolution efficiency can be increased in a controllable manner upon irradiation under visible light (>420 nm). So far, a majority of the reported polymeric photocatalysts are amorphous or semicrystalline systems without a long-range ordered arrangement in structures[3,4]. These are unfavorable for intermolecular charge transfer toward photocatalytic active sites. In this context, two-dimensional covalent organic frameworks (2D COFs), which constitute a typical class of crystalline organic porous materials[5–7], have attracted increasing attention toward the photocatalytic hydrogen evolution[8,9]. Such 2D COFs have a delocalized π-electronic system in the framework and layered structure that is stabilized by π-stacking. Therefore, it is rationalized that crystalline 2D COFs exhibit red-shifted absorption bands, enhanced exciton delocalization, and high charge mobility in a long range of crystalline domains.

To explore 2D COFs as photocatalysts, the molecular design is a predominant strategy. First of all, the construction of chemically stable linkages such as β-ketoenamine[10,11], sp² C=C bond[12,13], and triazine ring[14,15] for the COFs ensures the structural intactness under continuous irradiation. Some of them can further bind thoroughly in π-conjugated frameworks such as sp²-C COF[16–18] and covalent triazine frameworks[19,20], which can serve as light-harvesting antennae and electron relay stations to promote electron flow. Second, incorporation of electron-donating/withdrawing moieties such as diacetylene[21,22], hydrazone[23], azine[24], and sulfone groups[25] can adjust the redox capability and increase the exciton dissociation and charge transport. However, the studies on the effect of layered structures of 2D COFs have rarely been concerned with the photocatalytic H₂ evolution. It has been often observed that the X-ray diffraction signals of 2D COFs are significantly attenuated after photocatalysis[21,23], indicating that the stacking array of 2D COFs has been severely impaired or even amorphized. Thus, a weaker association between neighbouring layers induces a certain twist of the main backbones and limits the π-conjugation extension both in the frameworks and within layers. As the conjugation length of polymers is vital for the photogeneration of charge carriers, the structural distortion may compromise the photocatalytic performances[26]. Therefore, it is highly anticipated to develop accessible strategies that can preserve the layered π-stacking for 2D COFs.

Entrapping functional guests into porous structures has been demonstrated to be a flexible and effective strategy to modify the pore environment, most notably with linear polymers. In the pioneering studies, the polymer-threaded metal-organic frameworks have well-preserved structural stability and maintained accessible surface areas and opening pores, albeit with exposure to pressure[27], heat[28], or solvent treatment[29]. Meanwhile, the purposeful incorporation of functional polymers into pore channels can broaden functionality and enhance the performances of framework materials for a wide range of applications[30–35]. These inspiring studies motivate us to explore a polymer-infiltrated 2D COF with the aim to develop an innovative strategy for boosting the photocatalytic H₂ evolution. In this study, we synthesized a high-quality benzothiadiazole-containing COF (BT-COF) via a Schiff-base reaction and subsequent tautomerization in the presence of an organic base as a catalyst and modulator, forming a typical β-ketoenamine linkage. With a post-engineering modification, high-molecular-weight polyethylene glycol (PEG) was threaded in the 1D pore channels of BT-COF until the pore space

was almost stuffed. Since the BT-COF was tightly filled, the loss of the ordered stacking or even delamination was remarkably suppressed during the visible photocatalytic reaction. By strengthening the interlayered π-stacking, the PEG-filled BT-COF exhibits much enhanced performance for photochemical hydrogen evolution and structural stability over photocatalytic cycling. Furthermore, we demonstrate that this strategy is also applicable to different 2D COFs, elucidating its generality toward the development of COF-based photocatalysts.

## Results

**Synthesis of BT-COF.** As a high-crystallinity COF enforces the planarity of the whole 2D framework and accordingly facilitates the layered π-interaction, the synthesis was thus thoroughly optimized by varying the reaction conditions. A typical solvothermal reaction of 1,3,5-triformylphloroglucinol (Tp) with 4,4′-(benzo-2,1,3-thiadiazole-4,7-diyl)dianiline (BT) in a mixture of o-dichlorobenzene/n-BuOH (vol/vol, 19/1, 1 mL) in the presence of catalysts at 120 °C for 3 days generated the BT-COF in 63% isolated yield (Fig. 1a). Herein, pyrrolidine (Py) was used as a catalyst, instead of the aqueous acetic acid solution, for the formation of β-ketoenamine linkages (unless specified otherwise, the term BT-COF is only used for those synthesized with Py). As studied in our previous publication[36,37], Py has the benefit of the structural rearrangement through a Py-mediated transimination for the high-quality β-ketoenamine-linked COFs.

The chemical structure of the resulting BT-COF was characterized by various analytical methods. Fourier-transform infrared spectroscopy (FT-IR) spectra revealed that the characteristic vibration bands of C=O, C=C, and C–N bonds appeared at 1620, 1451, and 1255 cm⁻¹, respectively, which were derived from the formed β-ketoenamine subunits (Supplementary Fig. 1). Solid-state ¹³C cross-polarisation/magic angle spinning nuclear magnetic resonance (CP/MAS NMR) spectrum verified the presence of carbonyl C(a) at 185–187 ppm and secondary amine C(b) at 147 ppm, respectively, as well as the other aromatic signals corresponding to the phenyl C at 114–138 ppm and BT C (d) at 153 ppm (Supplementary Fig. 2). The chemical shift at 25 ppm was ascribed to alkane C(f) due to the terminal groups containing the tertiary amines arising from the reaction between Py and aldehydes[37]. By performing a local energy-dispersive X-ray spectroscopy, the elemental mappings disclosed that the distributions of C, N, O, and S atoms were overlapped in the same region, implying that the BT moiety uniformly distributed within the BT-COF (Supplementary Fig. 3). A close inspection at BT-COF by high-resolution transmission electron microscopy (HR TEM) revealed the presence of periodic structures (Fig. 1b).

The powder X-ray diffraction (PXRD) was employed to characterize the crystalline structure observed in the HR TEM image. There exhibited the remarkable diffraction peaks at 2.71°, 4.70°, 5.40°, 7.29°, and 26.15°, which were assigned to the (100), (110), (200), (210), and (001) facets, respectively (Fig. 1c). To interpret the lattice information, we calculated eclipsed (AA) and staggered (AB) stacking of the 2D single hexagonal layer with honeycomb topology in P6 and P6₃/m space groups, respectively (Supplementary Fig. 4). The AA-stacking model was more stable than the AB one, yielding an XRD pattern in a good agreement with the experimentally measured pattern. The Pawley refined PXRD pattern with the P6 space group gave the unit cell parameters of $a = b = 38.0430$ Å, $c = 3.4685$ Å, $\alpha = \beta = 90°$, and $\gamma = 120°$. The reproducible parameters are in line with the experimentally observed pattern with a negligible difference ($R_{wp} = 4.50\%$ and $R_P = 2.78\%$). The atomistic coordinates generated by calculations were summarized in Supplementary Table 1. Thus, as shown in Fig. 1d, the reconstruction of BT-COF

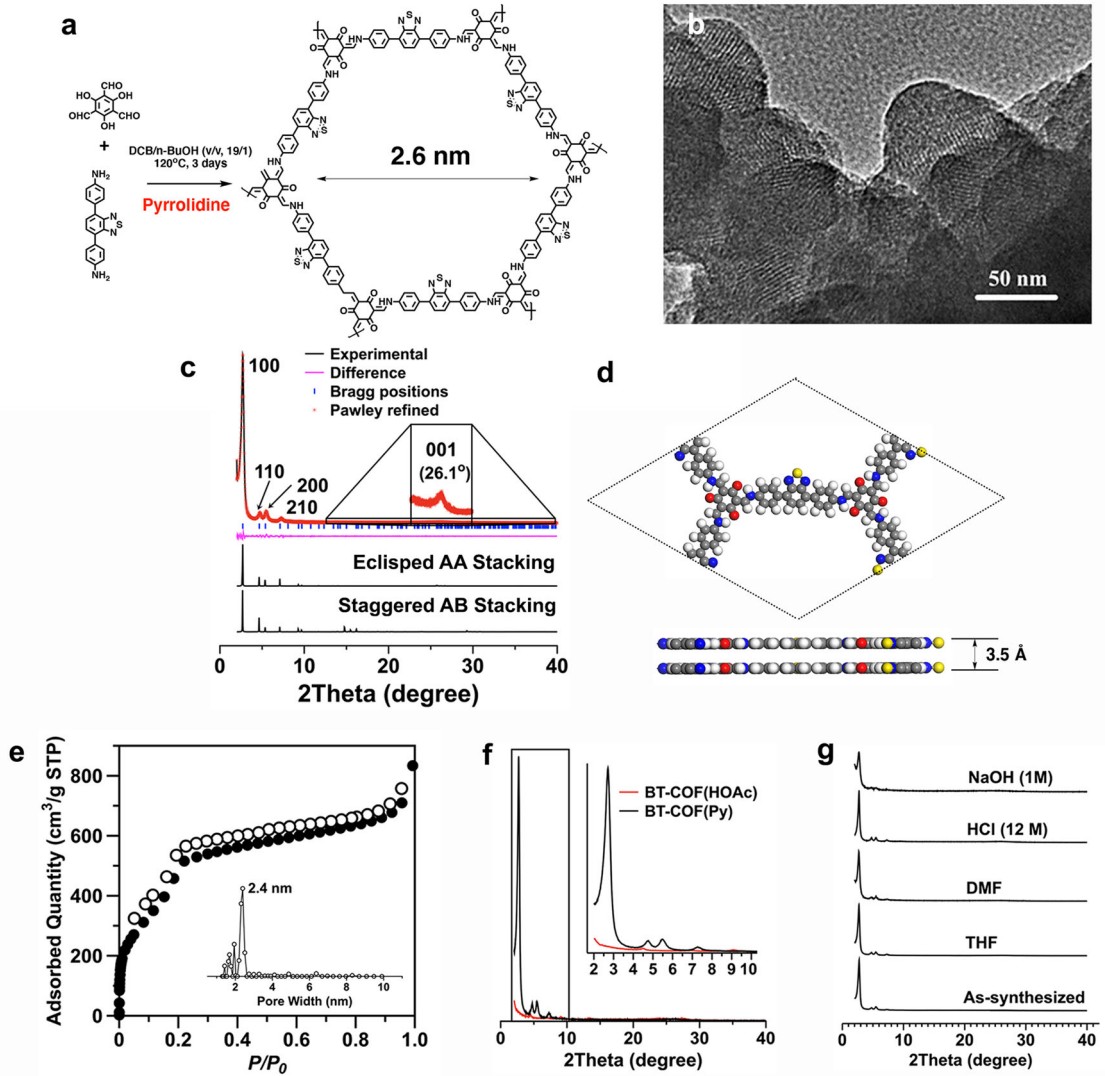

**Fig. 1 PXRD patterns, crystalline structure, TEM image, and porous properties. a** Synthesis of BT-COF by an aldimine reaction in a mixture of *o*-dichlorobenzene (DCB), *n*-butanol (*n*-BuOH), and pyrrolidine. **b** HR TEM image of BT-COF. **c** PXRD patterns of the experimentally observed (black) and Pawley refined (red) BT-COFs with their refinement difference (pink), and the simulated patterns for eclipsed AA- and staggered AB- stacking modes. **d** Lattice structure of BT-COF at top and side view (gray: carbon; white: hydrogen; blue: nitrogen; red: oxygen; yellow: sulfur). **e** Nitrogen adsorption (solid circle) and desorption (open circle) isotherm profiles and pore-size distribution (inset) of BT-COF obtained by nonlocal density functional theory modeling on the $N_2$ adsorption curve. **f** PXRD patterns of BT-COF synthesized with pyrrolidine (black curve) and 6 M HOAc aqueous solution (red curve) as catalysts, respectively. **g** Comparative crystalline stability of BT-COF in various solvents.

structure shows an extended hexagonal lattice with BT at the strut and Tp at the node. The presence of the (001) facet at 26.1°, corresponding to an even distance of 3.5 Å, suggests a π-stacked alignment in the *z*-direction perpendicular to the 2D sheets.

The porosity of BT-COF was assessed by nitrogen sorption measurement at 77 K. BT-COF featured a type IV sorption isotherm, indicative of a typical mesopore character (Fig. 1e). Brunauer–Emmett–Teller (BET) surface area and pore volume were calculated to be 1471 $m^2\,g^{-1}$ and 1.29 $cm^3\,g^{-1}$, respectively, both of which were ~80% of the theoretical values (1865 $m^2\,g^{-1}$ and 1.58 $cm^3\,g^{-1}$). Using the nonlocal density functional theory model, we found that the pore-size distribution was highly populated at 2.4 nm (Fig. 1e, inset). This is in line with the pore diameter of the crystal lattice revealed by the refined structure (2.6 nm). The amorphous analog poly(TpBT), which was synthesized by the reaction of Tp and BT under reflux in 1,2-dichlorobenzene, was found to be nonporous (Supplementary Fig. 5). Also, BT-COF(HOAc) catalyzed by HOAc aqueous

solution (6 M) gave a BET surface area of 65 $m^2\,g^{-1}$ and pore volume of 0.17 $cm^3\,g^{-1}$, indicating a disordering arrangement in structure (Supplementary Fig. 6). This was confirmed by the PXRD measurement as seen from Fig. 1f. Py is also effective for the synthesis of high-quality TP-COF (Supplementary Fig. 7), which resembles BT-COF, however, without the thiadiazole units.

For the estimation of crystalline stability, the BT-COF powders were immersed for 3 days in various solvents including tetrahydrofuran (THF), dimethylformamide (DMF), aqueous HCl (12 M), and aqueous NaOH solutions (1 M). The weight loss of the treated COFs was nearly ignorable and the solutions appeared colorless, indicating no decomposition inspected in such treatment (Supplementary Fig. 8). As observed in the PXRD patterns (Fig. 1g), the typical crystalline character has remained in most cases when compared with that of the pristine BT-COF. Nevertheless, the treatment with a basic aqueous solution led to a moderate decrease in X-ray diffraction intensity, inferring that the acidic condition is more suitable in photocatalysis than the basic one.

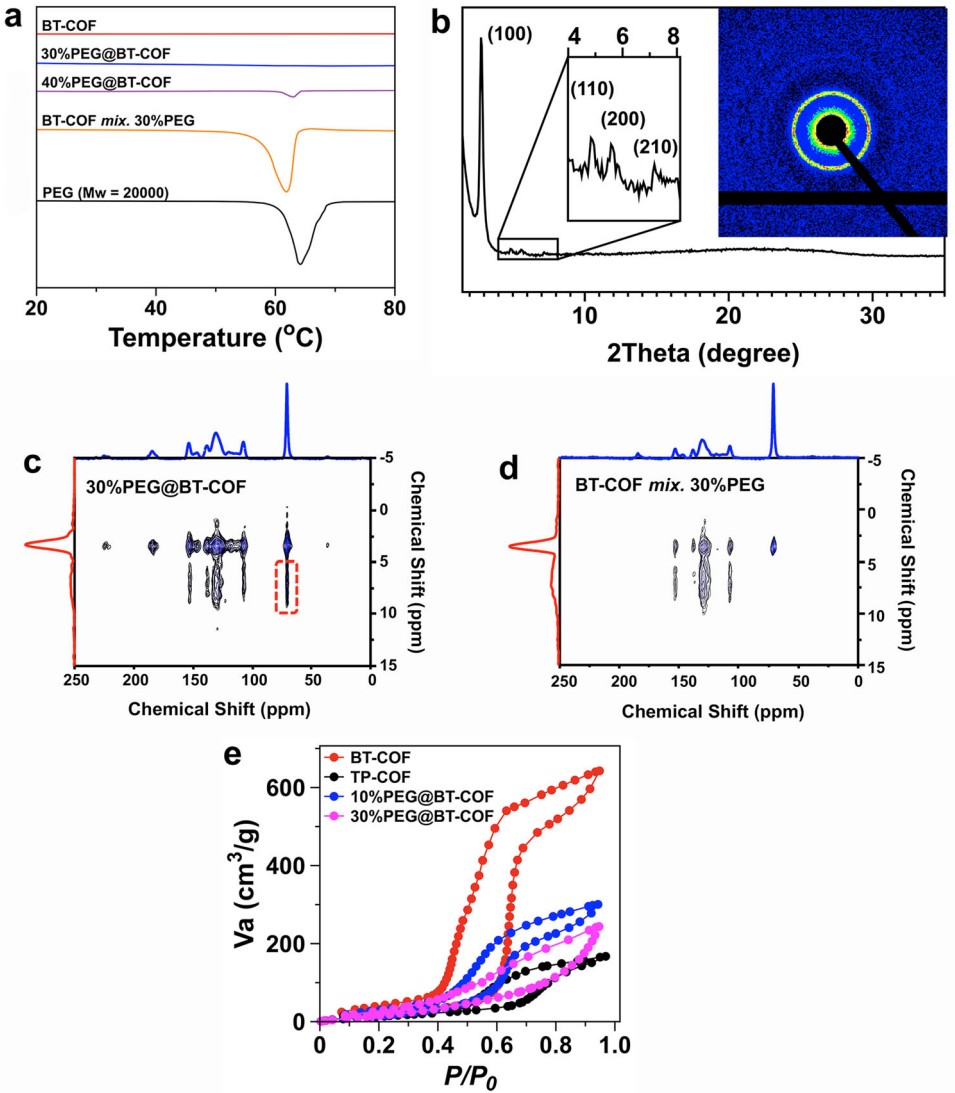

**Fig. 2 DSC profiles, NMR spectrum, WAXS pattern, and water vapor uptake profiles. a** DSC profiles of BT-COF, PEG, 30%PEG@BT-COF, 40%PEG@BT-COF, and the mixture of 30%PEG and BT-COF. **b** WAXS profile of 30%PEG@BT-COF (inset: two-dimensional WAXS pattern). **c, d** Solid-state 2D $^1$H-$^{13}$C HETCOR NMR spectra of **c** 30%PEG@BT-COF and **d** a mixture of BT-COF and 30 wt%PEG. **e** Water uptake profiles of BT-COF, TP-COF, 10%PEG@BT-COF, and 30%PEG@BT-COF at 298 K, respectively.

**PEG-stuffed mesopore channels in BT-COF**. An acetonitrile solution of PEG ($M_w$ = 20 kDa) with the two hydroxyl termini was mixed with BT-COF solids. Then, acetonitrile was evacuated progressively to allow thermal annealing, which proceeded at 100 °C for 12 h under reduced pressure. The operation implemented the low pressure-driven infiltration of the molten PEG within the 1D pore channels of COFs. To provide evidence of PEG insertion, differential scanning calorimetry (DSC) was employed to measure the crystallizing transition of PEG, which is normally present in a bulky state while is largely suppressed if present in nanopores due to the confinement effect[38]. As seen from Fig. 2a, the melting temperature of the bulk quantity of PEG is determined to be 64 °C in the DSC profile. When 30 wt% PEG relative to the total amount of BT-COF filled in pores, the melting temperature of PEG was not observed in the DSC heating curves. This is in sharp contrast to the simple mixture of PEG and BT-COF. With a further increase to 40 wt% PEG, a weak intensity peak corresponding to the melting temperature of PEG appeared, indicating that the pores were crammed and the excessive PEG chains crystallized on the periphery of BT-COF materials. Also, the thermogravimetric analysis

showed that the thermal decomposition of PEG chains appeared in the range from 350 to 450 °C, giving a corresponding weight loss of 9.1%, 16.6%, and 22.7%, respectively, which matches well with the filling of 10 wt%, 20 wt%, and 30 wt% PEG in the BT-COF (Supplementary Fig. 9).

The solid-state $^{13}$C CP/MAS NMR spectra of PEG@BT-COF verified the identical $^{13}$C resonances to the BT-COF, while a new peak emerging at 71 ppm could be assigned to ethoxy carbon on the PEG chains (Supplementary Fig. 10). The nitrogen sorption isotherms of PEG@BT-COF demonstrated a significant decrease in the adsorption capacity with the BET surface area of 223, 132, and 35 m$^2$ g$^{-1}$ for 10%PEG@BT-COF, 20%PEG@BT-COF, and 30%PEG@BT-COF, respectively (Supplementary Fig. 11). Also, the pore-size distributions were broadened and the dominant mesopore of ~2.4 nm disappeared when 30 wt% PEG was loaded (Supplementary Fig. 12). The porosity change may signify the presence of PEG chains within 1D pore channels. However, the structural periodicity of PEG@BT-COF cannot be precisely assessed from routine measurements such as PXRD (Supplementary Fig. 13) and TEM (Supplementary Fig. 14) as the

characterization of crystalline domains is challenging due to the presence of noncrystalline polymer chains. An in-depth investigation to acquire the precise stacking information of 2D frameworks was carried out using the wide-angle X-ray scattering (WAXS), which is a powerful tool to examine the regularity of molecular arrangement in polymer blends. As displayed in Fig. 2b, the dominated peaks at 2.79° corresponding to the (100) facets can be clearly detected in 30%PEG@BT-COF, as well as the other refined peaks ascribed to the (110), (200), and (210) planes. Once the PEG chains were extracted from 30%PEG@BT-COF with THF, the residual solids gave a high-intensity WAXS pattern containing all the peaks that could be indexed to the known hexagonal lattice of BT-COF (Supplementary Fig. 15). Meanwhile, the BET surface area rebounded to 1286 m$^2$ g$^{-1}$, which was close to the pristine COF (Supplementary Fig. 16).

To study the possibility of PEG adsorption on the crystallite domain boundaries, 30%PEG@BT-COF was sonicated in ethanol for 30 min and then examined by SEM. The samples retrieved before and after the sonication were essentially identical in terms of morphology and size (Supplementary Fig. 17). This agrees well with our assumption that PEG chains are embedded within pore channels so that the aggregates of COF microcrystals are preserved without evident dissociation. On the other hand, it rules out the possibility that the PEG chains are embedded into the interlayered space since the sandwiched PEG chains would have destroyed the ordered stacking and, in turn, exfoliated the layered structures. Also, as proved by quantum chemistry calculations, this would not be energetically favored (see Section 4.3 in Supplementary information for details).

Solid-state 2D $^1$H-$^{13}$C heteronuclear correlation NMR (HET-COR NMR) measurement was performed to directly prove that the infiltrated PEG chains were confined within pores. As displayed in Fig. 2c, the 2D HETCOR spectra of 30%PEG@BT-COF exhibit strong cross-peaks derived from the carbon resonance of PEG and the aromatic protons of BT-COF[39]. The result is caused by the intermolecular dipole interaction, strongly suggesting that the distance between PEG chains and BT-COF is very short[40]. In contrast, when 30 wt%PEG was merely mixed with BT-COF, no cross-correlation signals can be found in the 2D $^1$H-$^{13}$C HETCOR NMR spectra (Fig. 2d), indicating the distance between nuclei across the heterogeneous interface is larger than 1 nm. The findings directly elucidate that 30 wt% PEG chains should be predominately immobilized in the pore channels, which offer greatly large interfaces for interaction with PEG units.

Then, a quantitative analysis of structural durability was performed by calculating the average crystallite sizes with Scherrer equation, $\sigma = K\lambda/(\beta \cos \theta)$, wherein $K$ is a shape factor (assumed to be 1), $\beta$ is the full-width at half-maximum (FWHM), $\lambda$ is the X-ray wavelength, and $\theta$ is the Bragg angle of the diffraction peak. FWHM was obtained by fitting the strongest (100) X-ray scattering peak in the WAXS patterns (Supplementary Fig. 18 and Table 2)[41]. Compared with the parent BT-COF (44.6 nm), the crystalline domain size of 30%PEG@BT-COF was reduced to be 38.4 nm, preserving 86.1% of the initial size. Therefore, it is conclusively validated that filling the mesopores with long PEG chains can preserve a majority of the layered ordering stack for the BT-COF.

Based on dispersion-corrected density-functional theory (DFT) method, we found many possible binding positions for PEG oligomers residing on a single cut-out model of BT-COF (Supplementary Figs. 19 and 20). The binding energies between PEG oligomers and cut-out models linked by hydrogen bonding were calculated to give similar values. However, in the case of a dimer organized by two cut-out models, the binding energy of π-interaction is increased when PEG oligomer is positioned at the side of the dimer (Supplementary Figs. 21 and 22). It remarks that the attachment of PEG onto the pore wall of BT-COF can strengthen the interlayered π-interaction, and, accordingly, reduce the possibility of structural deformation (Supplementary Table 3). Also, with an increasing number of the loaded PEG chains, the layered structure of 2D COFs may be intrinsically fixed to avoid the offset between individual layers (Entry 4 in Supplementary Table 3).

Water vapor uptake performance was assessed at 298 K for the various COFs with or without the PEG loading. As displayed in Fig. 2e, BT-COF and TP-COF are both characteristic of the type II adsorption isotherms, of which BT-COF affords a better uptake capability of up to 643 cm$^3$ g$^{-1}$ that is equivalent to 52 wt% (STP) of water in a total mass. In contrast, TP-COF adsorbs only 14 wt% (STP) water under the same conditions. This is attributed to the number of polar heteroatoms including N, S, and O atoms as well as the higher surface area of BT-COF, thereby giving rise to the remarkable hydrophilicity for the pore environment. As the mesopores of BT-COF and TP-COF were filled up with high-molecular-weight PEG chains, the total water uptake significantly decreased, whereas the isotherms at low pressures still remained at a similar level (Fig. 2e and Supplementary Fig. 23), indicative of the preservation of hydrophilicity originating from the infiltrated PEG chains.

**DFT calculations**. The time-dependent DFT (TDDFT CAM-B3LYP) was used to simulate the ultraviolet (UV) spectra as well as optimize the geometry for the electronically excited state. We optimized several different cut-out models (G1–G3) to check convergence (Supplementary Fig. 24a). One can notice that the BT(G1) has a red-shifted peak at ~400 nm in comparison to that of the corresponding TP(G1) (360 nm). As more BT units are placed in the cut-out model by going from G1 to G3, the absorption at ca. 400 nm accordingly increases with a small redshift of a few nanometers (Supplementary Fig. 24b and Table 4). Then, the stacked counterpart (D1) was set up with the two G1 placed on top of one another with a stacking distance of 3.80 Å (Supplementary Fig. 24a). The oscillator strength of the peak at 400 nm doubles compared to the smallest model G1. This transition can be assigned mainly to the highest occupied molecular orbital-lowest unoccupied molecular orbital (HOMO-LUMO) excitation, which causes a large charge transfer from the delocalized HOMO, along with the Tp group, to the LUMO residing at the central BT unit (Supplementary Fig. 25). In contrast to the TP(G1), the absorption and the fluorescence spectra of BT(G1) are separated greatly (Supplementary Fig. 24c). The oscillator strength of the fluorescence transition from the first excited electronic state to the ground electronic state also decreases evidently from TP(G1) to BT(G1) (Supplementary Table 4). Using these values, we estimated the radiative decay for these COFs to be in the nanosecond time scale (see Sections 4.4 and 4.5 in Supplementary information for details). Exciton diffusion rate was estimated to be several orders of magnitude faster than this radiative decay rate in both BT-COF and TP-COF. Interestingly, our calculation shows that the out-of-plane diffusion rates for these π-stacked COF are higher than the in-plane rates, while the out-of-plane diffusion is very sensitive to the stacking and a slight tilt can decrease the rate by half.

To understand the electronic structure of the BT-COF and TP-COF, we performed DFT calculation on the cut-out cluster models to obtain the ionization potential (IP) as well as the electron affinity (EA) of the electronic ground and excited states (IP* and EA*). Following the method developed by Zwijnenburg and colleagues[42], we evaluated if these values straddle the required proton reduction potential and water

**Table 1 Calculated redox potentials for the different BT-COF and TP-COF cut-out models and those combined with water and diethyl ether, as displayed in the inset, respectively.**

| Model[a] | IP (V) | EA* (V) | EA (V) | IP* (V) |
|---|---|---|---|---|
| BT(G2) | 1.12 | 1.06 | −1.44 | −1.58 |
| BT(G1) | 1.03 | 1.03 | −1.49 | −1.49 |
| BT(G1)-C$_2$H$_5$OC$_2$H$_5$ | 1.04 | 0.88 | −1.49 | −1.33 |
| BT(G2)-H$_2$O | 1.07 | 1.03 | −1.48 | −1.43 |
| TP(G1) | 1.00 | 0.96 | −2.00 | −1.95 |

[a]The cut-out models are displayed as follows. The relative positions of diethyl ether and water with BT(G1) model were optimized and the interaction energy reached 10.30 and 9.48 kcal mol$^{-1}$, respectively.

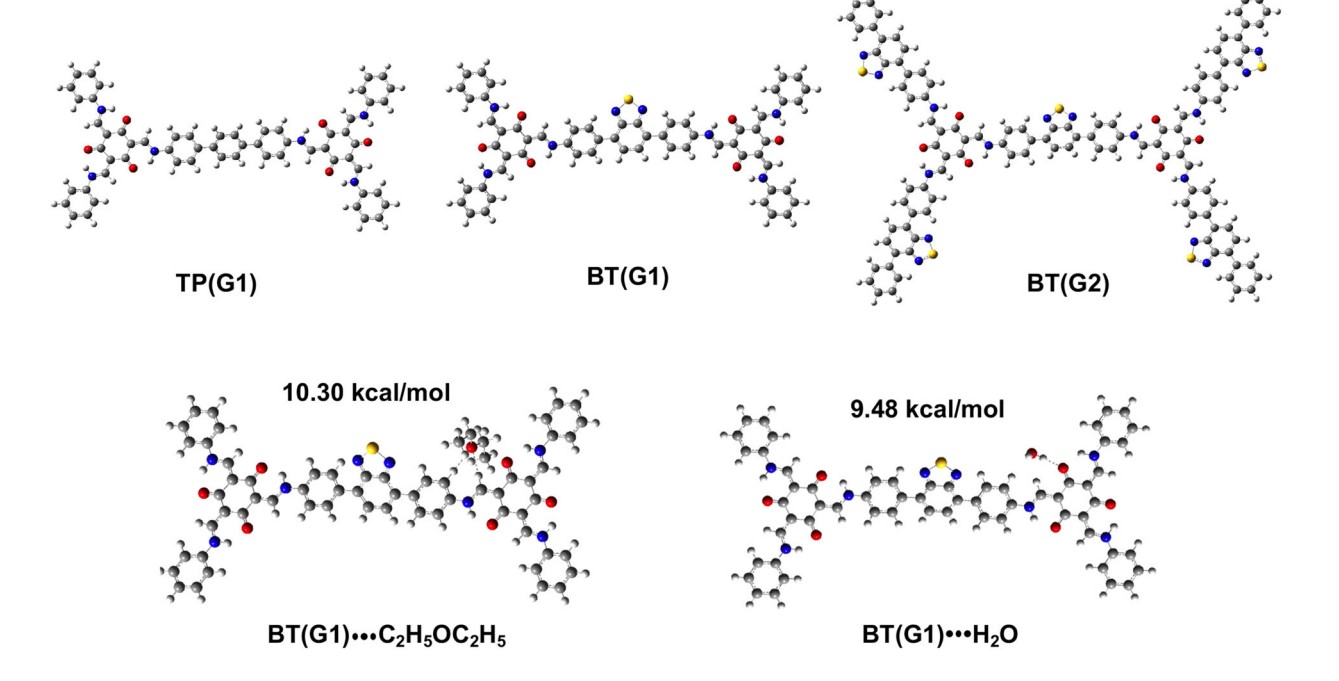

oxidation potential. Effect of water was treated using solvation model density[43]. Table 1 shows the calculated oxidation and reduction potentials against the SHE at pH = 0. From the comparison of the model sizes, the values for BT-COF seem to be converged with the smaller cut-out model. The EA and IP* calculated for the cut-out models of BT-COF and TP-COF have large negative values, implying the effective proton reducing ability displayed by these two COFs. On the other hand, IP and EA* are slightly below the water oxidation potential of 1.23 eV. As previously reported[25], the oxidation potentials of ascorbic acid (AA) are slightly negative compared to that of water oxidation; thus, they can be used as a sacrificial electron donor. Indeed, our calculated values for the one- and two-hole oxidation potentials are <1 V (see Supplementary Table 5). We further evaluated the effect of the interactions between the cut-out models with water as well as PEG by attaching a water molecule and a diethyl ether (C$_2$H$_5$OC$_2$H$_5$) molecule as the PEG representative. The obtained potentials show very little variation, <0.1 V (Table 1). Conclusively, the electronic state of BT-COF does not change even when interacting with PEG.

**Photocatalytic hydrogen evolution.** The UV–visible (UV-vis) reflectance spectra of the various COFs were measured in the solid state to access the bandgap structures. To elucidate the

unique photophysical properties based on the BT-COF, the analogs of amorphous poly(TpBT) and TP-COF were adopted as control groups. As displayed in Fig. 3a, the absorption onsets were found at 532, 574, 579, 593, and 599 nm for the TP-COF, poly(TpBT), BT-COF(HOAc), BT-COF, and 30%PEG@BT-COF, respectively. BT-COF exhibits an evident redshift in the onset, possibly due to the extended π-conjugation given by the layered structure. The two analogs, i.e., BT-COF(HOAc) and poly(TpBT), have a ca. 15 nm blueshifted onset compared to BT-COF. When 30 wt% PEG was loaded in the BT-COF, the absorption spectrum remained in the visible region with a maximum at 413 nm. The onset of absorption was very similar to that of the BT-COF, indicating the minimal effect of PEG on the electronic structure of BT-COF, agreeing well with the calculation results.

The optical bandgaps were calculated by using Tauc plots. BT-COF and 30%PEG@BT-COF in the solid state gave similar values of 2.09 and 2.07 eV, respectively (Supplementary Fig. 26). Both values lie in a suitable range of bandgaps for visible photocatalytic H$_2$ evolution. With the identical measurement conditions, all of the analogs including TP-COF (2.33 eV), BT-COF(HOAc) (2.15 eV), and poly(TpBT) (2.16 eV) offered the larger optical bandgaps. A more elaborate resolution of energy band structure was performed by a synchrotron radiation photoemission spectroscopy under vacuum (Supplementary Fig. 27). As shown in Fig. 3b, the valence bands of BT-COFs with and without 30 wt

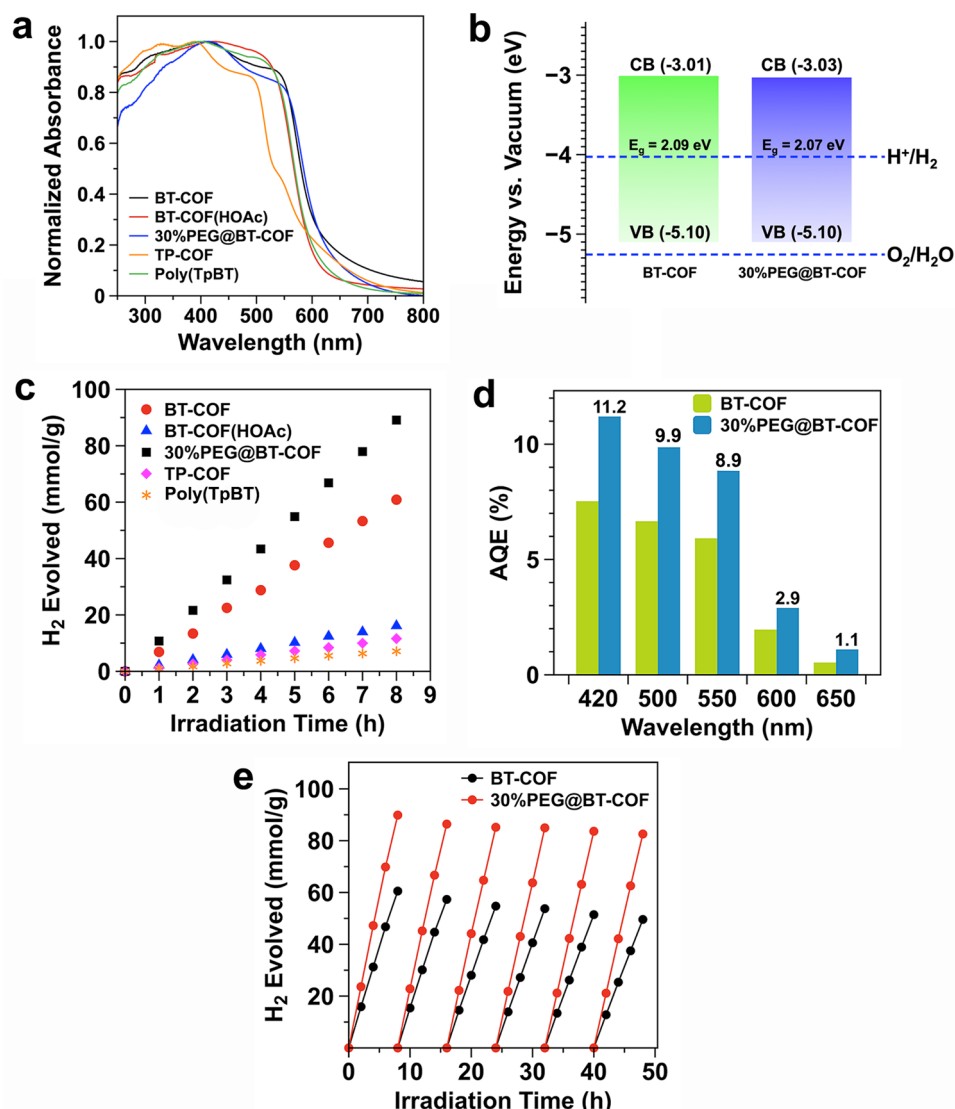

**Fig. 3 Absorption spectra, band structure, and photocatalytic H₂ evolution. a** UV − vis diffuse reflectance spectra of poly(TpBT), BT-COF, BT-COF (HOAc), 30%PEG@BT-COF, and TP-COF measured in solid state. **b** Energy band structures of BT-COF and 30%PEG@BT-COF determined by synchrotron radiation photoemission spectroscopy. **c** Time course for photocatalytic H₂ production under visible irradiation for BT-COF, BT-COF(HOAc), 30% PEG@BT-COF, poly(TpBT), and TP-COF, respectively. **d** Apparent quantum efficiency of BT-COF and 30%PEG@BT-COF at five different incident light wavelengths. **e** Photocatalytic H₂ production with the BT-COF and 30%PEG@BT-COF, respectively, upon 48-h illumination of visible light.

% PEG were finely measured and both lie at $-5.10$ eV versus vacuum level. Using the aforementioned optical bandgaps, the conduction bands were calculated to be $-3.01$ and $-3.03$ eV for the BT-COF and 30%PEG@BT-COF, respectively. As the deviation of their band structures is negligible, it is thus inferred that 30%PEG@BT-COF has a similar electronic structure for photocatalytic hydrogen evolution.

Photocatalytic activity of hydrogen evolution was investigated by a slightly modified method from the earlier studies[25]. The reaction was conducted in a top-irradiated vessel containing COF grains (10 mg) loaded with Pt (5 wt% of COF) in water (100 mL) and in the presence of 0.1 M AA. Figure 3c shows the amount of H₂ produced as a function of the photoirradiation time under visible light (>420 nm) for different photocatalytic materials, all of which remained active and provided a steady rate of H₂ evolution over 8 h of irradiation. The average hydrogen evolution rates (HERs) were calculated to be 1.45 mmol g$^{-1}$ h$^{-1}$ for TP-COF, 7.70 mmol g$^{-1}$ h$^{-1}$ for BT-COF, and 11.14 mmol g$^{-1}$ h$^{-1}$ for 30%PEG@BT-COF. In contrast, the amorphous poly(TpBT) and

ill-crystalline BT-COF(HOAc) showed much lower photocatalytic activity, giving the HERs of 0.8 and 2.02 mmol g$^{-1}$ h$^{-1}$, respectively.

Next, the effect of reaction conditions on photocatalytic performances has been thoroughly studied. When the pH of the system was adjusted from 1.6 to 5.6 (Supplementary Fig. 28), the highest HER was up to 8.44 mmol g$^{-1}$ h$^{-1}$ at pH = 3.6, which verified that the hydrogen source was not derived from AA since its p$Ka_1$ is at 4.1. Without Pt cocatalyst in the presence of AA donor, BT-COF alone could evolve H₂ at a constant rate of 0.25 mmol g$^{-1}$ h$^{-1}$. By the in situ photoreduction of different amounts of Pt precursors (H₂PtCl₆), the HERs reached a maximum of 7.95 mmol g$^{-1}$ h$^{-1}$ using 7 wt% Pt and then declined (Supplementary Fig. 29). Given that similar activity was seen by 5 and 7 wt% of Pt, 5 wt% Pt was adopted for all the photocatalytic tests. In such a case, the quantitative analysis performed by inductively coupled plasma could confirm 3.54 wt% Pt loaded in BT-COF and 3.73 wt% in 30%PEG@BT-COF (Supplementary Table 6). HR TEM images displayed a uniform

deposition of Pt nanoparticles, while a close inspection at the size distribution revealed that the 30%PEG@BT-COF resulted in the dominated size of 2.3 nm, which was a little larger than that (1.9 nm) of Pt nanoparticles produced within the BT-COF (Supplementary Fig. 30). This is probably due to the infiltrated PEG that provides a number of nucleation sites for the favorable growth of Pt nanoparticles[44] and subsequently the broadened size distributions. A similar increase in the average size of Pt nanoparticles (2.2 nm) was observed when the short PEG chains ($M_w = 2000$) were threaded in the BT-COF (Supplementary Fig. 30). Thus, it is likely that more Pt nanoparticles are deposited on the surface of the PEG@BT-COF instead of in the pores. This feature should be beneficial for enhancing the photocatalytic performance. Nevertheless, the HER of 30%PEG-20k@BT-COF was much larger than those of the BT-COFs infiltrated with short PEG-2000 and PEG-400 chains (Supplementary Fig. 31). Therefore, the slight change of the deposition sites or sizes of Pt nanoparticles may not be the decisive factors responsible for the improvement in the photocatalytic performance of the 30%PEG-20k@BT-COF because the increasing sizes of Pt nanoparticles usually impair the photocatalytic activity[45].

Apart from the effect of pH and Pt nanoparticles, we found that the HERs could elevate from 8.90 to 11.14 mol g$^{-1}$ h$^{-1}$ when the amount of PEG ($M_w = 20k$ Da) was increased from 10 to 30 wt% (Supplementary Fig. 32). When the short-chain PEG molecules ($M_w = 400$ or 2000) were loaded in BT-COFs under identical conditions, they could be released into the aqueous solutions from the nanopores during photocatalytic tests due to relatively small molecular sizes and better hydrophilicity. This is evidenced by the FT-IR spectra of the 48-h recycled PEG@BT-COF as nearly all characteristic bands of ethylene oxides corresponding to PEG were absent (Supplementary Fig. 33). Hence, the recycled COFs infiltrated with a short PEG chain exhibited weak X-ray scattering signals similar to those of the parent BT-COF (Supplementary Fig. 34). Thus, implying again that long-chain PEG plays a pivotal role in stabilizing COF structures during photocatalysis.

To investigate the wavelength-dependent photocatalytic activity, the apparent quantum efficiency (AQE) was evaluated by comparing the incident photonic numbers with the number of electrons used for H$_2$ production. As shown in Fig. 3d, the AQE values are 11.2%, 9.9%, 8.9%, 2.9%, and 1.1% at 420, 500, 550, 600, and 650 nm for the 30%PEG@BT-COF, respectively, all of which are nearly 1.5-fold higher than those of BT-COF accordingly. To the best of our knowledge, the reported AQE at 420 nm is one of the highest values ranked among the top COF-based photocatalysts used in hydrogen evolution (Supplementary Table 7).

A long-term photocatalytic activity for H$_2$ production was then accessed for BT-COF and PEG@BT-COF, respectively, under identical conditions. As displayed in Fig. 3e, the linear correlation between the produced H$_2$ quantity and the irradiation time could be retained for each cycle upon 48-h visible irradiation in the presence of 30%PEG@BT-COF. Furthermore, the 8-h accumulated production of H$_2$ only declined by 8% over the six cycles. In contrast, BT-COF gave a continuous decline in the production of H$_2$ with a total decrease of 21% in 48 h. As evidenced by FT-IR measurement, the BT-COF and PEG@BT-PEG retrieved from the 48-h photocatalysis cycling both retained their chemical integrity due to the stable β-ketoenamine linkages even in acidic solutions (Supplementary Fig. 35). Also, the HR TEM image exhibited a discrete distribution of Pt nanoparticles on these COFs without evident aggregation or disjunction (Supplementary Fig. 36). However, the crystallinity of the recycled BT-COF was largely deteriorated, indicated by the weak and broad PXRD profiles as well as the drastic decrease in surface area (195 m$^2$ g$^{-1}$)

(Supplementary Fig. 37). This is probably due to the severe loss of 2D layered structures, as also reported in the precedent literatures[21,23]. For the 30%PEG@BT-COF collected after the 48-h cycles, the PXRD measurement showed no distinctive peaks due to the PEG filling up in the periodic mesopores (Supplementary Fig. 38). After the PEG was extracted from the pores, the typical crystalline pattern of BT-COF was recovered (Supplementary Fig. 39). When the recycled BT-COF was subjected to the same solvent extraction, the low crystalline structure was unchanged (Supplementary Fig. 40). By using the Scherrer equation, we found that the average crystallite sizes of PEG@BT-COF slightly decreased from 38.4 to 33.1 nm before and after the photocatalytic cycles, maintaining 86.2% of the initial domain size (Supplementary Fig. 18 and Table 2). In sharp contrast, there was merely a half of crystallite size (48.4%) remaining in the recycled BT-COF (21.6 nm) under the identical conditions of photocatalytic cycling. Therefore, the ordered stacking structure is effectively maintained in the PEG@BT-COF during photocatalysis. The photogenerated charge transport may occur between the proximity layers accordingly.

Then, we carefully investigated the origin of crystallinity loss for the BT-COF during photocatalysis. As displayed in Fig. 4a, the photo-deposition of Pt nanoparticles on BT-COF has already imposed a significant effect on the PXRD pattern of BT-COF. The predominated diffraction peak at 2.7° is noticeably attenuated in comparison to the parent one. Without the addition of the Pt precursors, the XRD signal intensity did not vary since the sample was only exposed to the UV–vis light in 0.1 M AA (Supplementary Fig. 41). In view of the stable β-ketoenamine linkages, we reason that the interlayered force of BT-COF is not strong enough to remain the ordered stacking structure in the photoreduction reaction (Fig. 4a).

After the linear PEG chains are incorporated into the pore channels, the random coils are reasonably stretched in the constrained space, although a fully extended conformation is not thermodynamically allowed as the minimum of entropy loss must be ensured during the conformation transition. To confirm our speculation on the change of the PEG conformation, a dissipative particle dynamics (DPD) simulation was performed to investigate the universal natures of polymer chains in a coarse-graining route. PEG was modeled by DPD beads sequentially linked by harmonic bond and the water solvent was modeled as single beads. The hard repulsive Lennard–Jones potential was applied to implement the constraint of the nanotube to the PEG and solvent beads. Moreover, hydrogen bonding was added between the PEG beads and the nanotube wall. As displayed in Fig. 4a, the radius of gyration ($R_g = 5.0$ nm) for the random coil in the bulk estimated by DPD simulations is similar to that of PEG-20k in the θ-condition[46]. By a quantitative comparison, $R_g$ of the PEG-20k model in the bulk is ~4 times the radius of the nanotube. Consequently, the PEG chains in the nanotube have to be elongated and partially adhered to the pore walls via hydrogen bonding. When the radius of the nanotube is expanded, the stressed PEG chain is relaxed to some extent but still retains the elongated state (Supplementary Fig. 42). Therefore, once a number of pore channels of BT-COF were threaded with the elongated PEG coils, the interlayered dislocation can be effectively suppressed to maintain the coaxial stacking. Indeed, with an increase in PEG amounts from 10 to 30 wt%, the more densely filling of pore space made the ordering layered structure more stable, leading to the HERs rising from 8.90 to 11.14 mol g$^{-1}$ h$^{-1}$. Analogously, when the pores of TP-COF were filled up with PEG, the HER of 30%PEG@TP-COF was nearly two times higher than that of the parent TP-COF, reaching 2.82 mmol g$^{-1}$ h$^{-1}$ (Supplementary Fig. 43). After the long-term irradiation cycling, there was only 5.9 wt% of PEG chains (with respect to the total

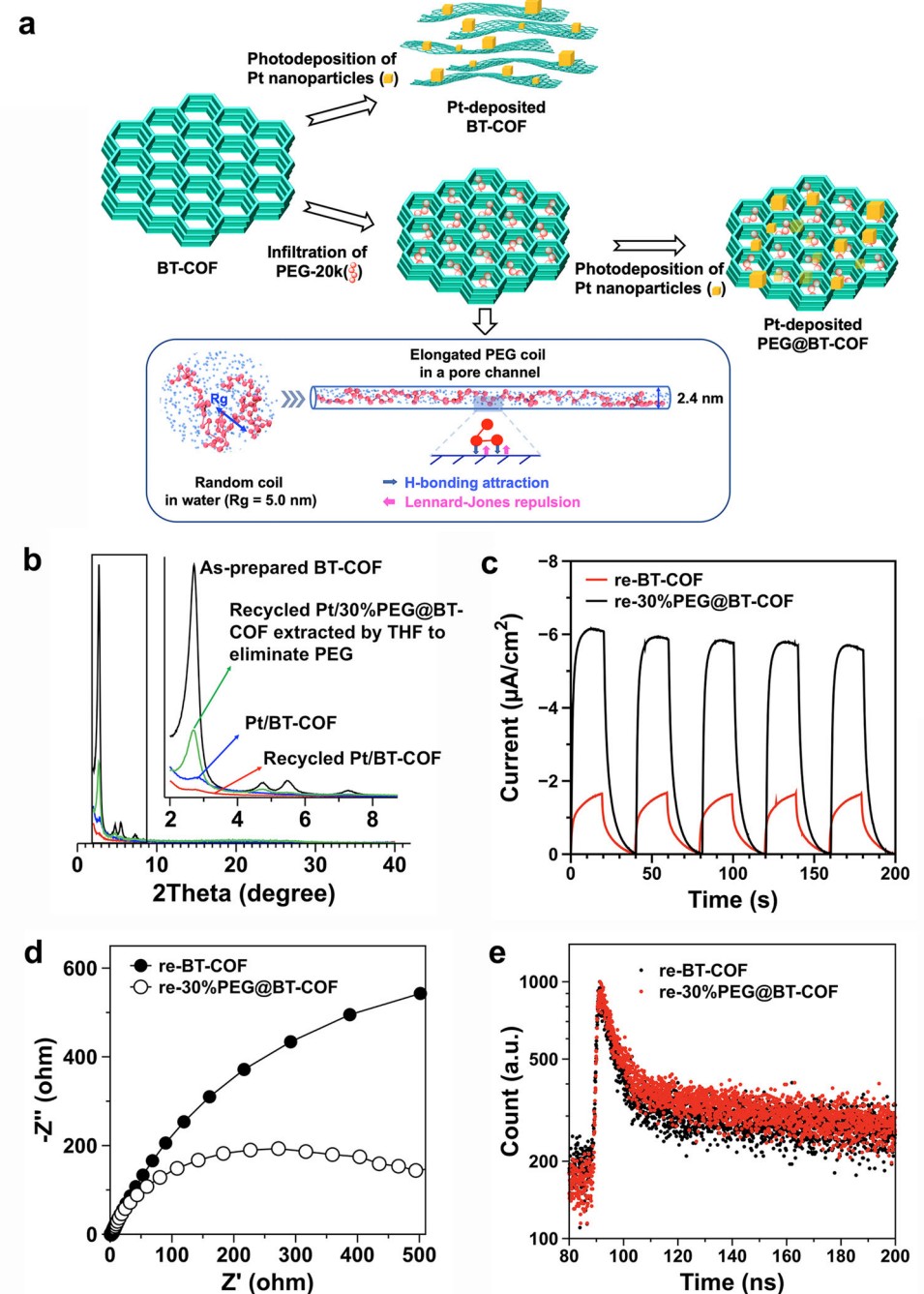

**Fig. 4 PXRD patterns, proposed mechanism, and photophysical measurements. a** Illustration of structural transformation of COF and PEG@COF during the deposition of Pt cocatalyst. PEG conformations in water and in a pore channel (2.4 nm) are simulated by the dissipative particle dynamics (DPD) method in the canonical ensemble, respectively. According to the coarse-graining model, harmonic-bond-linked DPD beads and individual beads are applied to model PEG and water, respectively. **b** PXRD patterns of the as-prepared BT-COF (black curve), the Pt-deposited BT-COF before (blue curve), and after (red curve) 48-h cycling and the recycled 30%PEG@BT-COF (green curve) after 48-h photocatalysis, of which the loaded PEG was washed off by THF prior to the PXRD measurement. **c** Transient photocurrent responses, **d** Nyquist plots, and **e** TCSPC measured fluorescence decay profiles of the re-BT-COF and re-30%PEG@BT-COF obtained after 48-h photocatalysis, respectively.

mass of the composite) leaking out from the composite 30% PEG@BT-COF (Supplementary Fig. 44). This ensures a sustainable $H_2$ evolution at a steady rate during the long-term photocatalysis.

With all of the findings in mind, we compiled the HER data of the different photocatalysts in Table 2. The amorphous poly (TpBT) gives the lowest HER due to the absence of a typical 2D COF structure (Entry 1). The positive role of PEG-stabilized

layered structure is remarkable on the photocatalytic $H_2$ evolution both for BT-COF (Entries 2 and 3) and TP-COF (Entries 4 and 5). As a control, when PEG was merely mixed with the BT-COF dispersed in water, the HER was similar to that of BT-COF (Entry 6 and Supplementary Fig. 45), indicating that freely dispersed PEG in solution has negligible effect. In addition, considering the earlier reports that alcohols such as methanol, ethanol, and isopropanol can serve as sacrificial electron donors[47,48], we tested

**Table 2 Correlation of hydrogen evolution rates (HERs) with the stacking structures of COFs.**

| Entry | Sample | Stacking state[a] | HER (mmol h$^{-1}$ g$^{-1}$)[c] |
|---|---|---|---|
| 1 | Poly(TpBT) | Amorphous | 0.89 |
| 2 | BT-COF | Low ordering | 7.70 |
| 3 | 30%PEG@BT-COF | PEG-stabilized | 11.14 |
| 4 | TP-COF | Low ordering | 1.45 |
| 5 | 30%PEG@TP-COF | PEG-stabilized | 2.82 |
| 6 | 30%PEG mix. BT-COF[b] | Low ordering | 7.60 |

[a]The stacking state signifies the nature of layered structures during the photochemical hydrogen evolution.
[b]30wt% of PEG was mixed with the BT-COF for the photoreduction reaction.
[c]All HERs measured using the same instruments, optical setup, and reaction conditions: 10 mg sample, 3.5 ± 0.2 wt% Pt, 0.1 M ascorbic acid, 100 mL water, 300 W Xe light source equipped with λ > 420-nm cut-off filter. HERs were obtained from the average 8-h photoirradiation and normalized to the sample mass.

if PEG can act as a donor by removing the AA donor while keeping other conditions constant. One can see that only a trace of H$_2$ was produced in the presence of 30%PEG@BT-COF (Supplementary Fig. 46). This strongly suggests that PEG is not able to serve as a sacrificial electron donor toward the photocatalysis reaction.

**Photophysical mechanism study**. Owing to the possible change of layered structures before and after photocatalysis in water, a series of photophysical measurements have been thoroughly conducted to get insight into the impact. The UV–vis spectra showed that the optical bandgaps were consistent before and after the 48-h irradiation cycling for both BT-COF and 30%PEG@BT-COF (Supplementary Figs. 47 and 48). Upon switching the photoirradiation (>420 nm) on and off, both the recycled COFs quickly responded to the incident light and generated a large photocurrent. However, the recycled 30%PEG@BT-COF had a much higher transient photocurrent than the recycled BT-COF (Fig. 4c). Meanwhile, the electrochemical impedance spectroscopy reflected the same tendency that the resistance of the recycled 30%PEG@BT-COF was much lower than the recycled BT-COF (Fig. 4d). The excited-state lifetime of the recycled sample was measured in the solid state by the time-correlated single-photon counting (Fig. 4e). The average exciton lifetime of the recycled 30%PEG@BT-COF was estimated to be 25.5 ns, longer than that of the recycled BT-COF (17.6 ns). Prior to photocatalysis, all of the photophysical characteristics were almost the same for the as-prepared BT-COF and 30%PEG@BT-COF without photodeposition of Pt nanoparticles (Supplementary Fig. 49). Therefore, it is reasoned that the prolonging exciton lifetime and enhanced charge transport may be a result of the ordered π-stacking maintained in the recycled PEG@BT-COF.

**Discussion**
In this study, we have proposed a general and feasible strategy to maintain the ordering layered structure of 2D COF during photocatalysis. This, in turn, elevates the performance of 2D COF photocatalysts in the hydrogen evolution reaction. The designed framework incorporates BT in the strut as a photosensitizer, while it maintains its chemical stability by the β-ketoenamine linkage. Using Py as a catalyst for the aldimine reaction, high crystallinity and superior porosity are achieved for the resulting BT-COF. The coaxially ordered stacking renders molecular planarity, extends the π-electronic conjugation, and tunes the optical bandgap for

great visible harvesting. By using a post-engineering method, the high-molecular-weight PEG chains are stuffed into the 1D mesopore channels of BT-COF and are firmly anchored on the pore wall by hydrogen bonding. The effect of PEG filling suppresses the gliding of 2D neighbouring layers and accordingly stabilizes the desirable layered structure. This favors the photogenerated charge transfer and prolongs the exciton lifetime for the subsequent proton reduction. Without the PEG stuffing in pores, the layered AA-stacking of BT-COF is readily re-arranged into a low-ordering assembly when Pt cocatalysts are deposited in the photoreduction. This may imply that the interaction between Pt, water, and photogenerated polarons on the COF skeletons offsets the interlayered forces. In particular to the β-ketoenamine-linked COFs that are typically composed of tri(N-salicylidenea-niline) at the node[10,11], a number of polar groups including C=O and C-NH make the stacking structures vulnerable to external stimuli[49]. Therefore, with the stuffing of PEG linear chains in 1D pore channels, the ordered π-stacking within layers is effectively stabilized and in turn, exhibits a much better photocatalytic H$_2$ evolution activity (11.14 mmol g$^{-1}$ h$^{-1}$) than the parent COF (7.70 mmol g$^{-1}$ h$^{-1}$), with the steady cycling performances and the maximum AQE (11.2% at 420 nm). The strategy is also effective for the other COF photocatalysts as seen for TP-COF. Therefore, given that the COF/polymer assembling strategy is readily accessible and controllable, the configuration of commonly used photocatalytic systems such as donor–acceptor heterojunction and Z-scheme could be diversely explored on the COFs' platform.

## Methods
**Synthesis of BT-COF**. A Pyrex tube was charged with Tp (12.6 mg, 0.06 mmol), BT (28.7 mg, 0.09 mmol), and a mixture of o-dichlorobenzene (0.95 mL) and n-butanol (0.05 mL). Py (0.1 mL) or aqueous acetic acid (0.1 mL, 6 M) was then added into the tube as a catalyst. The mixture was sonicated for 5 min and degassed using three freeze–pump–thaw cycles. The tube was then sealed and kept in an oven at 120 °C for 3 days. The precipitate was filtered off, exhaustively washed by Soxhlet extractions with tetrahydrofuran for 3 days, and dried at 120 °C under vacuum for 12 h to yield BT-COF as deep red powders in 63% isolation yield (24 mg).

**Synthesis of PEG-stuffed COFs**. The infiltration of PEG chains into the COF channels was carried out by the low pressure-driven method. Typically, a known amount of activated (i.e., guest-free) COFs was suspended in anhydrous acetonitrile. The acetonitrile solution of PEG ($M_w$ = 20 kDa) was then added and the mixture was evacuated progressively under reduced pressure (0.3 kPa). After solvent elimination, thermal annealing was allowed to proceed at 100 °C for 12 h under reduced pressure to drive the insertion of PEG chains into pore channels of COFs.

**Electronic structure computations**. For the different cluster cut-outs, we performed geometry optimization for the neutral, anion, and cation states using the Coulomb-attenuating method Becke's three-parameter hybrid (CAM-B3LYP) functional[50]. The excited electronic state of the neutral molecule was calculated using the time-dependent variant. All calculations were performed with the Gaussian09 program[51].

**Photocatalytic hydrogen evolution**. Catalysts (10 mg), a certain amount of hexachloroplatinic acid and 0.1 M aqueous solution of AA (100 mL), were charged in a hermetic device mainly composed of a quartz tube and sealing components. The resulting suspension was ultrasonicated for 10 min. The reaction mixture was purged with Ar flow to remove air and illuminated with a 300 W Xe lamp (100 mW cm$^{-2}$, Perfect Light PLS-SXE 300). A cut-off filter (Kenko L-42) was used to achieve visible-light irradiation (>420 nm). The temperature of the reaction solution was kept at 10 °C by a flow of cooling water. The amount of H$_2$ evolved was determined using gas chromatography (GC-7890A with thermal conductivity detector (TCD) and Ar carrier, Agilent or SHIMADZU GC-2014 with a TCD detector). The HERs were determined from a linear regression fit. After the photocatalytic experiments, the photocatalysts were recovered by washing with water, THF, and acetone before drying at 120 °C.

## Data availability
All the data that support the findings of this study are available from the corresponding author upon reasonable request.

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

## Acknowledgements
This work is supported by the National Natural Science Foundation of China (Grants Nos. 51973039, 21774023, and 51633001) and STCSM (18520744800). K.T. acknowledges support from Academia Sinica (CDA-106-M05). L.W. thanks for the support of the China Postdoctoral Science Foundation (BX20200317).

## Author contributions
J.G., H.X., and K.T. conceived the project. T.Z. performed the experiments. J.G. and T.Z. wrote the manuscript. X.H., R.W., and H.Z. assisted in the experiments and characterizations. L.W. and H.X. conducted the photocatalytic hydrogen evolution and photophysical measurements. K.T. performed the DFT calculations. Q.Y. conducted the crystal lattice simulations. Q.S. and W.L. simulated the polymer conformations. K.T., J.U., H.X., L.W., and C.W. assisted in the manuscript preparation. All authors participated in discussions.

## Competing interests

The authors declare no competing interests.
