## [Peer Review File · Nature Communications]

REVIEWER COMMENTS

Reviewer #1 (Remarks to the Author):

In the article entitled with “PEG-Stabilized Coaxial Stacking of Two-Dimensional Covalent Organic Frameworks for Enhanced Photocatalytic Water Splitting”, authors have very nicely explored a general and feasible strategy to maintain the ordering of layered structure of 2D COF during photocatalysis by introducing PEG chains into the 1D mesopore channels of BT-COF. This modification of 2D COF results in the enhanced HER evolution during photocatalysis. I am glad to recommend the publication of this paper in Nature Communication once the authors consider the following points:

(i) In page 10, authors have mentioned that with the decrease of oscillator strength of the fluorescence transition from first excited electronic state to the ground electronic state, electron-hole recombination by light emission is possibly slow. Please add some references to support it and also if possible the authors are suggested to calculate the electron-hole recombination rate for TP(G1) and BT(G1).

(ii) In the same page (p-10), authors have calculated IP/IP^* and EA/EA^* of cut-out cluster models of BT-COF and TP-COF by following the method of Zwijnenburg et al. Authors are suggested to mention how they calculated IP/IP^* and EA/EA^* in electronic structure calculations in supporting information and how they relate these quantities with redox potentials.

(iii) “While EA^* (~1.6 V) is greater than the water oxidation potential of 1.23 V, the IP (1.05 V) is just slightly below it, which means that these COFs require sacrificial electron donor to assist in the oxidation reaction” –Please explain it more clearly and also give some references to support it.

Reviewer #2 (Remarks to the Author):

In this work, Jia Guo et al. reported the encapsulation of polyethylene glycol (PEG) in 2D covalent organic frameworks (COFs), which could suppress loss of their ordered stacking during photoreduction reaction. As a result, the nanocomposites between COFs and PEG exhibited higher photocatalytic activity compared with the original COFs because of the effective charge carrier transport as well as prolonged exciton lifetime. However, this manuscript did not bring the sense of high novelty that is usually expected from a report in Nature Communications because similar concepts for the structural stabilization of porous materials by polymer insertion have been already

reported in many papers (A. Celeste, et al., *J. Am. Chem. Soc.*, 2020, 142, 15012; B. Le Ouay, et al., *J. Am. Chem. Soc.*, 2019, 37, 14549; L. Peng, et al., *J. Am. Chem. Soc.*, 2019, 141, 12397). In addition, the manuscript essentially lacks analysis and characterizations to support the claims made. Therefore, I cannot recommend the publication of this manuscript in *Nature Communications*.

1. The authors should provide direct evidences for the infiltration of the polymers in the COFs. I would recommend the authors to measure multinuclear 2-D NMR that is one of the most powerful techniques to confirm the encapsulation of guest species inside porous materials. When polymer chains are incorporated into pores and are in close proximity to the host materials at the molecular level, cross-correlations are observed because of heteronuclear magnetization transfer between polymer chains and hosts (P. Duan, et al., *J. Am. Chem. Soc.*, 2019, 141, 7589).

2. I am not convinced of the authors' claim that insertion of PEG into COF nanopores leads to stabilize the structure. The authors should investigate how the long chain PEG can increase the stability of the COFs. I am afraid that low molecular weight PEG or even small molecules could stabilize the structure of the COFs. Accommodation of guest species would stabilize the structure of the COFs regardless of the molecular size.

3. The authors should fully characterize the conformation of the PEG chains in the COFs. It is still suspicious that the polymer chains would form an extended conformation along the 1-D channel direction. Note that PEG chain is very flexible and even could form globular structure in mesopores of COFs.

4. Considering the small domain size of COFs (Fig. 1d), the PEG chains are hardly encapsulated with a fully extended conformation in the COF channels, as was shown in Fig 4b.

5. The distance between the neighboring layers in the COF is 3.5 Å, as confirmed by PXRD patterns (Fig. 1b), which is comparable with the thickness of the PEG chains. Therefore, side diffusion of the polymer chains beyond the 1D channels could be also allowed. In this regard, again, the current data cannot support the threading of the polymer chains along the channel direction.

6. The catalytic activity of Pt nanoparticles would be affected by their particle size and location (inside or outside the pores). In this regard, it should be better to provide the detailed characterizations of the Pt nanoparticle and investigate effects of PEG on the growth of Pt nanoparticles.

7. What is the peak at 187 ppm in ¹³C CP-MAS NMR spectrum of BT-COF (Fig. S2)?

8. Bo Wang et al. has already reported hybridization of PEG and COF (Bo Wang, et al., *J. Am. Chem. Soc.*, 141, 1923). This work is highly relevant to the present work and should be cited.

9. Stacking structure of the COFs might be retained after the introduction of the PEG

chains into the COFs, as confirmed by the wide-angle X-ray scattering measurement (WAXS) measurements (Fig. 2). However, I am not still convinced that the crystalline structures of the COFs were completely maintained, as shown in Fig. 4b. There is no direct evidence for the ordering along the (110) and (210) planes.

10. The authors should perform quantitative analysis for the degree of structural maintenance upon the insertion of the polymer chains. Is it possible to discuss the crystallinity using WAXS?

Reviewer #3 (Remarks to the Author):

The authors describe the structural stabilization of 2D COFs by incorporating PEG chains into the pores. The authors claim that this threading of polymer chains into the pores is stabilizing the ordered arrangement of the stacking layers, especially under photocatalytic conditions. Indeed the PEG infiltrated COFs show higher and more stable photocatalytic hydrogen evolution from water. The COF and COF-PEG composites are analyzed with a variety of methods to explain the effect of the PEG infiltration.

I think this is an interesting addition to the fast growing field of photocatalysis with COFs, even though the improvement by adding PEGs is not such pronounced. As the authors show, small variations in Pt content or pH already have a more distinct effect on the hydrogen evolution yield than the PEG threading.

Anyway, because of the high current interest in COFs and photocatalytic HER, the work can be suitable for Nature Communication.

There are just some minor concerns:

1. In general, the authors have chosen to combine many schemes and measurements in one figure. This results in very small diagrams. Just as example all XRD patterns in Figure 1 g can be hardly recognized, and if the “crystalline structure is retained without any signs of destruction” is thus hard to judge. Some measurements should be moved to the SI or, if possible, shown their again in higher resolution.
2. For the stability tests, not just the XRD pattern before and after treatment should be compared but also the sample weight. If some amount of the COF is getting decomposed and dissolved, this is not necessarily seen in the PXRD of the residue.
3. That the “thermogravimetric analysis proved again that 30wt% PEG or less was loaded in BT-COF” is a quite unspecific statement, especially as values are given in the figure captio of Figure S8. However, as there is no clear step for PEG decomposition

in the TGA, these loading values should be handled with care, especially to give numbers with one digit seem not trustworthy. Which “theoretical values” are meant in the figure caption of Figure S8?

4. The authors should describe the difference between their PXRD and WAXS measurements. Especially as the “WAXS” measurement show also very small angles, thus this seem not to be a conventional WAXS.

5. In both solid state NMR measurements there is an obvious peak at 25 ppm (plus some smaller ones at this area), which are certainly no spinning side bands. How can these peaks assigned?

6. The increasing stability of layer stacking is sacrificed by an overall lower surface area of the COF. Thus is the apparent surface area of a COF photocatalyst not such important as it is claimed in recent literature?

7. Even though the absolute values are not such high, the effect of PEG threading on HER seem to be more pronounced for TP-COF than for BT-COF. Can this be due to an increased hydrophilicity of TP-COF when PEG chains are introduced? At least in water sorption measurement TP-COF seem to have low water uptake, a water sorption measurement of PEG@TP-COF would be therefore interesting.

REVIEWER COMMENTS:

Reviewer #1:

In the article entitled with “PEG-Stabilized Coaxial Stacking of Two-Dimensional Covalent Organic Frameworks for Enhanced Photocatalytic Water Splitting”, authors have very nicely explored a general and feasible strategy to maintain the ordering of layered structure of 2D COF during photocatalysis by introducing PEG chains into the 1D mesopore channels of BT-COF. This modification of 2D COF results in the enhanced HER evolution during photocatalysis. I am glad to recommend the publication of this paper in Nature Communication once the authors consider the following points:

1. In page 10, authors have mentioned that with the decrease of oscillator strength of the fluorescence transition from first excited electronic state to the ground electronic state, electron-hole recombination by light emission is possibly slow. Please add some references to support it and also if possible the authors are suggested to calculate the electron-hole recombination rate for TP(G1) and BT(G1).

Response: Thanks for the kind suggestion of the reviewer.

In the revised manuscript, we have removed this sentence which is not giving clear numbers. Instead, we have used the calculation results to estimate the spontaneous emission lifetime of the BT-COF and TP-COF. As given in Supplementary Information, we show that the BT-COF has a lifetime that is ~ 4 times larger than TP-COF, but both are in the nanosecond order. In addition, we have added the two new sections in Supplementary Information, estimating the energy transfer rates. We have used a simple model based on the energy differences of the electronic transition energies to estimate the transfer rates. We also find that the in-plane and out-of-plane transfer rates are comparable and that the stacking form greatly affects the out-of-plane transfer rates.

The aforementioned revisions have been made in the main text, *Page 11, Line 18-22*, and *Page 12, Line 1-2*, “Using these values, we estimated the radiative decay for these COFs to be in the nanosecond time scale (see Section 4.4 and 4.5 in Supplementary Information for details). Exciton diffusion rate was estimated to be several orders of magnitude faster than this radiative decay rate in both BT- and TP-COF. Interestingly our calculation shows that the diffusion rates in-plane and out-of-plane for these π -stacked COF are similar. In addition, the out-of-plane diffusion is very sensitive to the stacking and a slight tilt can decrease the rate by half. The calculation details were shown in *Supplementary Information, Page S11-S13, Section 4.4 and 4.5*, including:

4.4 Theoretical estimation of excited-state lifetime

Using our cluster cutout model, we estimated the excited state lifetime and the rate for exciton dissociation in the present BT-COF and TP-COF system. Using Einstein’s

coefficient of spontaneous emission:^[S17]

$$A(s^{-1}) = \frac{1}{4\pi\epsilon_0} \frac{64\pi^4\nu^3}{3hc^3} \mu_{ij} = \frac{2\pi e^2}{\epsilon_0 m_e c \lambda^2} f_{ij} = 6.67 \times 10^{13} (nm^2 s^{-1}) \frac{f_{ij}}{\lambda^2}$$

where, ϵ_0 is the vacuum permittivity, h is the Planck's constant, c is the speed of light, ν is the frequency of the emission, μ_{ij} is the transition dipole moment between states i and j . The second equation is defined using the oscillator strength f_{ij} , electron charge e , the mass of an electron m_e , and the emission light wavelength λ . We obtain the third equation from the constants' numerical value, where the wavelength is given in nm, and the oscillator strength has no units. Using this equation and the fluorescence peak position and oscillator strength given in Supplementary Table 8, we estimate the spontaneous emission rate to be $3.3 \times 10^8 s^{-1}$ and $1.4 \times 10^9 s^{-1}$ for BT and TP, respectively. For these COFs, this rough estimate places the lifetime due to the excited state emission to be in ns time scale.

4.5 Theoretical estimation of excited state energy transfer rates

For the energy transfer rate or the exciton diffusion rate, we will assume localized excitations and use the formula

$$k_{ET}(s^{-1}) = \frac{2\pi}{\hbar} |J_{ij}|^2 \int \tilde{\sigma}_{Em}(E) \tilde{\sigma}_{Ab}(E) dE$$

where, J_{ij} is the electron coupling between the two monomer cutouts, and $\tilde{\sigma}_{Em/Ab}(E)$ are the energy dependence of the normalized emission and absorption spectrum. Similar equations have been used for the energy migration in porphyrin-based metal-organic frameworks.^[S18] The key question here will be how much faster the k_{ED} will be in comparison to the spontaneous emission rate mentioned above. Here we will be modeling the exciton diffusion BT-COF(G1)+BT-COF(G1*) \rightarrow BT-COF(G1*)+BT-COF(G1), where the electronically excited energy is transferred between neighboring BT monomer units. There are many different ways to obtain the electron coupling, such as using the transition dipole, i.e., the Forster transfer model, or atomic transition densities, or constraint density functional theory.^[S18, S19] Here, we use a dimer model of the G1 cutout model and performed TD-CAMB3LYP calculation to obtain the electronic transition energies. Then half of the energy difference between the 1st and 2nd electronic transitions will be used to estimate the electronic coupling. Here, we note that the 1st and 2nd electronic transitions for the dimer are the symmetric and antisymmetric combination of the monomer excitation. For the in-plane transfer, we will use a linked G1 monomer unit, while for the out-of-plane transition, we will use the stacked dimer (Supplementary Fig. 49). Upon geometry optimization, the stacked dimer complex showed a slight tilting, which is consistent with previous reports on 2D-COF, which mention that AA stacking is stable, and the perfectly eclipsed form is not the global minima.^[S20, S21] Thus, the two G1 cut out are not perfectly overlapping along the stacking axis. We used the calculated absorption and emission spectra given in Supplementary Fig. 24 to approximate the overlap integral.

Supplementary Fig. 49 Schematic geometry of the in plane and stacked dimer model of BT-COF(G1)

As given in Supplementary Table 8, our estimate of the energy transfer rate, k_{ET} , of the photoexcited state for the TP- and BT-COF gives values which are several orders faster than the spontaneous emission rate. We can see that the difference in BT and TP-COF's spectral overlap is why we see a difference in the energy transfer rate. An important aspect of the calculation is that if one looks at the in-plane versus out-of-plane electron coupling, we see that in our model, the out-of-plane values are slightly larger for both COFs. We also evaluated the situation of the perfect overlap. As given in Supplementary Table 8, the coupling becomes twice when the BT-COFs are perfectly stacked, thereby showing how sensitive the electron coupling is toward the stacking geometry. Before ending, we must consider possible errors from our simulation. First, our spectral overlap has been obtained from simplified theoretical spectra rather than experimental results. However, we think the relative shift in peak position for the absorption and emission spectra is reproduced. We do not think the error will be several orders of magnitude, making k_{ET} reach ns time scale.

Supplementary Table 8. Electron coupling J_{ij} , the spectral overlap, and an estimate of the electron transfer rate for G1 BT-COF and TP-COF calculated using TD-CAMB3LYP/def2SVP. Values in parenthesis is obtained from perfect overlapping stacked dimer calculations.

	J_{ij} (eV)	Overlap (eV ⁻¹)	k_{ET} (s ⁻¹)
BT-COF (out of plane)	0.058 (0.11)	0.019	1.3×10^{12}
BT-COF (in plane)	0.033	0.019	2.0×10^{11}
TP-COF (out of plane)	0.056	0.266	1.8×10^{13}
TP-COF (in plane)	0.047	0.266	5.6×10^{12}

We do note that recent studies have experimentally shown that the exciton diffusion may not be a step-by-step hopping as assumed here, ^[S19] and domain sizes can affect

the exciton diffusion.^[S22] Therefore, further studies are needed to clarify the transport mechanism fully, but it is beyond the scope of the paper. The methods used here to estimate the rates of emission and energy transfer rely on very simplified models. However, we think the general trend between the TP- and BT-COF can be obtained from these models. On a final note, we mention that recent studies have shown exciton dynamics in ps order, consistent with our estimate given in Table 8.^[S22,S23]

2. In the same page (p-10), authors have calculated IP/IP* and EA/EA* of cut-out cluster models of BT-COF and TP-COF by following the method of Zwijnenburg et al. Authors are suggested to mention how they calculated IP/IP* and EA/EA* in electronic structure calculations in supporting information and how they relate these quantities with redox potentials.

Response: According to the reviewer, we have clarified the electronic structure calculations and the relation with redox potentials in *Supplementary Information, Page S8, Line 1-19*, “We optimized the neutral geometry singlet ground electronic state for all the cluster cutout models using CAMB3LYP/defSVP. For the optimization of the cation and anion geometries in the doublet ground electronic state, we used the unrestricted variant of CAMB3LYP. To calculate the free energy’s entropic contribution at 298 K, we used harmonic approximation for the vibrational motion. On the other hand, we have ignored the rotational and translational contribution since we use a cluster cutout model that will be immobile in the actual COF geometry. This part is slightly different from the method developed by Zwijnenburg and co-workers,^[S14,S15] but only results in a change of 0.03 eV at most. Taking the free energy differences of the neutral and cation states, we obtained the adiabatic ionization potential (IP). On the other hand, the electron affinity (EA) is obtained from the neutral and anion states’ energy difference. Using TD-CAMB3LYP, we optimized the geometry on the electronic state corresponding to the strong absorption in the UV spectra given in Supplementary Fig. 24. Taking the free energy difference between the neutral ground state and this electronic excited state, we obtain the adiabatic excitation energy (EX). Using these values, we obtain the ionization potential of the excited state (IP*) by IP-EX, while the electron affinity of the excited state (EA*) is given by EA+EX. We used the IUPAC recommended 4.44 V for the standard hydrogen electrode absolute potential.”

3. “While EA* (~1.6 V) is greater than the water oxidation potential of 1.23 V, the IP (1.05 V) is just slightly below it, which means that these COFs require sacrificial electron donor to assist in the oxidation reaction” –Please explain it more clearly and also give some references to support it.

Response: It was found that, in the previous version, we were using the vertical excitation energy rather than the adiabatic transition energy. Therefore, the potential values have changed in **Table 1**. However, we see that the EA* and IP are still below the water oxidation potential. Therefore, we have rewritten the corresponding sentence

in the main text, *Page 12, Line 12-16*, “On the other hand, IP and EA* are slightly below the water oxidation potential of 1.23 eV. As previously reported²⁵, the oxidation potentials of ascorbic acid are slightly negative compared to that of water oxidation; thus, they can be used as a sacrificial electron donor. Indeed, our calculated values for the one- and two-hole oxidation potentials are less than 1 V (see Supplementary Table 5).”

Table 1. Calculated redox potentials (V) for the different BT-COF and TP-COF cut-out models and those combined with water and diethyl ether, as displayed in Inset, respectively.

	BT(G2)	BT(G1)	BT(G1)- C ₂ H ₅ OC ₂ H ₅	BT(G2)- H ₂ O	TP(G1)
IP	1.12	1.03	1.04	1.07	1.00
EA*	1.06	1.03	0.88	1.03	0.96
EA	-1.44	-1.49	-1.49	-1.48	-2.00
IP*	-1.58	-1.49	-1.33	-1.43	-1.95

In addition, we have performed the calculation on the one- and two-hole oxidation potential of ascorbic acid using the present level of theory. We report this in *Supplementary Information, Page S8-S9*, “We used the same quantum chemistry method, CAMB3LYP-D3/def2SVP with SMD, to estimate the one- and two-hole oxidation potential of ascorbic acid (H₂A). In principle, we only need to calculate ascorbate radical (HA) by removing one hydrogen atom, and dehydroascorbic acid (DHA) by removing two hydrogen atoms. However, in the aqueous phase DHA is known to react with water to form the bicyclic diol form (DHAD). Following the study by Tu et al.,^[S16] we calculated the cluster of ascorbic acid with two water molecules (H₂A...H₄O₂), the cluster of ascorbate radical with two water molecules (HA...H₄O₂), dehydroascorbic acid with two water molecules (DHA...H₄O₂), and the bicyclic diol form of DHA with one water molecule (DHAD...H₂O) (Supplementary Fig. 48). Using the free energies based on the harmonic approximation for the vibrational modes of these clusters, we obtain the oxidation potential with respect to the standard hydrogen electrode. To validate the accuracy of the present calculation method to estimate the oxidation potential, we also calculated the oxidation potential for water using the same quantum chemistry methods by calculating H₂, O₂, and H₂O. We find that the present method can give reasonable estimates for the water oxidation potential. As given in Supplementary Table 5, we confirmed that the calculated one- and two-hole oxidation potential for ascorbic acid is negative compared to the calculated water oxidation potential.”

Supplementary Fig. 48 Schematic geometries of ascorbic acid with two water molecules ($H_2A \dots H_4O_2$), the cluster of ascorbate radical with two water molecules ($HA \dots H_4O_2$), dehydroascorbic acid with two water molecules ($DHA \dots H_4O_2$), and the bicyclic diol form of DHA with one water molecule ($DHAD \dots H_2O$).

Supplementary Table 5. The calculated potential with respect to standard hydrogen electrode at pH=0 for the oxidation reactions of ascorbic acid and water calculated using CAMB3LYP-D/def2SVP SMD. The geometries of the hydrated clusters are given in Figure 44.

	Potential (V)
$HA \dots H_2O + H^+ + e^- \rightarrow H_2A \dots H_4O_2$	0.88
$DHAD \dots H_2O + H^+ + e^- \rightarrow HA \dots H_4O_2$	-0.51
$DHAD \dots H_2O + 2H^+ + 2e^- \rightarrow H_2A \dots H_4O_2$	0.37
$\frac{1}{4}O_2 + H^+ + e^- \rightarrow \frac{1}{2}H_2O$	1.17

Reviewer #2:

In this work, Jia Guo et al. reported the encapsulation of polyethylene glycol (PEG) in 2D covalent organic frameworks (COFs), which could suppress loss of their ordered stacking during photoreduction reaction. As a result, the nanocomposites between COFs and PEG exhibited higher photocatalytic activity compared with the original COFs because of the effective charge carrier transport as well as prolonged exciton lifetime. However, this manuscript did not bring the sense of high novelty that is usually expected from a report in Nature Communications because similar concepts for the structural stabilization of porous materials by polymer insertion have been already reported in many papers (A. Celeste, et al., J. Am. Chem. Soc., 2020, 142, 15012; B. Le Ouay, et al., J. Am. Chem. Soc., 2019, 37, 14549; L. Peng, et al., J. Am. Chem. Soc., 2019, 141, 12397). In addition, the manuscript essentially lacks analysis and characterizations to support the claims made. Therefore, I cannot recommend the publication of this manuscript in Nature Communications.

Response: We are very grateful to the reviewer for his/her time reading this manuscript and providing critical comments. In this revised manuscript, we have carefully considered the comments and concerns raised by the reviewer and made revisions accordingly. We believe that our study is meaningful and will generate broad interest to the research fields related to physical chemistry, catalysis, materials, and polymer chemistry based on the following reasons.

(1) Our study provides solid experimental and computational results to demonstrate that polymer-infiltration strategy is able to efficiently stabilize the layered structure of 2D COFs during photocatalysis. As pointed out by the reviewer, the polymer-infiltration strategy has been reported for stabilizing MOFs. However, the previous studies primarily focus on the structural stability of MOFs upon applying physical stimuli such as pressure. Whether the polymer-infiltration strategy is effective in stabilizing porous materials towards a catalytic reaction is still not known. We believe that our study provides significant insights into the structural stability of COFs during photocatalysis and offers new perspectives in developing porous materials for a wide range of catalytic applications via this polymer-infiltration strategy.

(2) This polymer-infiltration strategy is established not only as a general and feasible methodology to retain the ordered structure of 2D COFs but also facilitate exciton diffusion and charge transport in COFs. Consequently, incorporating PEG chains into COFs dramatically enhance the photocatalytic performances in addition to provide structural stability. To the best of our knowledge, this unique polymer-infiltration enabled photophysical effect has not been discovered in previous studies, which would generate a profound impact for future development of intrinsically unstable materials towards various applications.

(3) To convincingly prove that the enhanced stability and photocatalytic performance indeed originate from the introduction of PEG chains into the COFs, various

characterizations including WAXS patterns, theoretical calculations, N₂ adsorption-desorption isotherms, DSC measurements, photocatalytic tests, and the newly conducted 2D ¹H-¹³C NMR measurements have been carried out in our investigation. Moreover, control experiments further confirm that PEG chains are able to stabilize BT-COF and promote charge separation during photocatalysis. Thus, we believe that our characterizations are now comprehensive and the structure-property relationship is evident.

In this revised manuscript, we have supplemented a large amount of additional data to further underpin our claims according to the comments and suggestions provided by the reviewer. Our findings provide new prospects towards design and synthesis of highly active and stable organic photocatalysts toward solar-to-chemical energy conversion.

1. The authors should provide direct evidences for the infiltration of the polymers in the COFs. I would recommend the authors to measure multinuclear 2-D NMR that is one of the most powerful techniques to confirm the encapsulation of guest species inside porous materials. When polymer chains are incorporated into pore in close proximity to the host materials at the molecular level, cross-correlations are observed because of heteronuclear magnetization transfer between polymer chains and hosts (P. Duan, et al., *J. Am. Chem. Soc.*, 2019, 141, 7589).

Response: Thanks for the valuable suggestion of the reviewer. According to the suggestion, we collected the two-dimensional (2D) ¹H-¹³C heteronuclear correlation (HetCor) spectra for 30%PEG@BT-COF and physically mixed sample (30%PEG *mix.* BT-COF), respectively. The corresponding results are included in the revised manuscript, Fig. 2c and 2d. The measured 2D ¹H-¹³C NMR spectra of 30%PEG@BT-COF exhibits the strong cross peaks derived from the carbon resonance of PEG and aromatic protons of BT-COF. The cross peaks produced by the intermolecular dipole interaction suggest that the distance between PEG and BT-COF is in a very short range as reported (Ref. 35. P. Duan *et al. J. Am. Chem. Soc.*, 2019, 141, 7589 and Ref. 36 P. Sozzani, *et al. Nat. Mater.* 2016, 5, 545.). In contrast, when PEG was only physically mixed with BT-COF, no cross-correlation signals can be found in the 2D ¹H-¹³C HetCor NMR spectra, indicating the distance between nuclei across the heterogeneous interface is large (>1 nm). Thus, this additional measurement further confirm that the infiltrated PEG chains are confined in the pore channels of BT-COF.

The corresponding revision has been made in the main text, **Page 9, Line 10-19, “Solid-state 2D ¹H-¹³C HETCOR NMR measurement was performed to directly prove that the infiltrated PEG chains were confined within pores. As displayed in Fig. 2c, the 2D heteronuclear correlation (HetCor) spectra of 30%PEG@BT-COF exhibit strong cross peaks derived from the carbon resonance of PEG and aromatic protons of BT-COF³⁵. The result is caused by the intermolecular dipole interaction, strongly suggesting that the distance between PEG chains and BT-COF is very short³⁶. In contrast, when 30wt.%PEG was merely mixed with BT-**

COF, no cross-correlation signals can be found in the 2D ^1H - ^{13}C HetCor NMR spectra (Fig. 2d), indicating the distance between nuclei across the heterogeneous interface is larger than 1 nm. The findings directly elucidate that 30wt.% PEG chains should be predominately immobilized in the pore channels, which offer the greatly large interfaces for interaction with PEG units.”

Fig.2c Solid-state 2D ^1H - ^{13}C HETCOR NMR spectra of (c) 30%PEG@BT-COF and (d) a mixture of BT-COF and 30wt.%PEG.

2. I am not convinced of the authors' claim that insertion of PEG into COF nanopores leads to stabilize the structure. The authors should investigate how the long chain PEG can increase the stability of the COFs. I am afraid that low molecular weight PEG or even small molecules could stabilize the structure of the COFs. Accommodation of guest species would stabilize the structure of the COFs regardless of the molecular size.

Response: Thanks for the issue addressed by the reviewer. According to the concern, we further prepared the two control samples by infiltrating low molecular weight PEG (i.e. $M_w = 400$ or 2000 Da) into BT-COF (hereafter referred to as 30%PEG-400@BT-COF and 30%PEG-2000@BT-COF). We performed the WAXS characterization to investigate whether the short-chain PEG could stabilize the structure of the BT-COF during the photocatalysis. As shown in **Supplementary Fig. 33**, the recycled 30%PEG-2000@BT-COF and 30%PEG-400@BT-COF after photocatalytic tests only exhibits one broad (100) peak of BT-COF with the weakened intensity and increased peak width. This finding directly proves that the short-chain (low molecular weight) PEG cannot stabilize the structure of BT-COF during photocatalysis. Meanwhile, no distinctive peaks corresponding to PEG can be found in the FT-IR spectra of the recycled 30%PEG-2000@BT-COF and 30%PEG-400@BT-COF (**Supplementary Fig. 32**). This is in sharp contrast to that of 30%PEG-20K@BT-COF. This phenomenon reveals that the low-molecular-weight PEG chains were released from the BT-COF during photocatalysis. Consequently, low-molecular-weight PEG or small molecules cannot be used as structural stabilizers for COFs in terms of photocatalytic applications.

The corresponding revision has been made in the context, *Page 15, Line 6-14*, “When the short-chain PEG molecules (e.g. $M_w = 400$ or 2000) were loaded in BT-COFs under identical conditions, they could be released into the aqueous solutions from the nanopores during photocatalytic tests due to relatively small molecular sizes and better hydrophilicity. This is evidenced by the FT-IR spectra of the 48-h recycled PEG@BT-COF as nearly all characteristic bands of ethylene oxides corresponding to PEG were absent (Supplementary Fig. 32). Hence, the recycled COFs infiltrated with short-chain PEG exhibited the weak X-ray scattering signals, as similar as those of the parent BT-COF (Supplementary Fig. 33). Thus, implying again that long-chain PEG plays a pivotal role in stabilizing COF structures during photocatalysis.”

Supplementary Fig. 32 FT IR spectra of the recycled PEG20k@BT-COF, PEG400@BT-COF and PEG2000@BT-COF after 48-h photocatalysis cycles, respectively.

Supplementary Fig. 33 WAXS profiles of the recycled BT-COF, PEG400@BT-COF and PEG2000@BT-COF after 48-h photocatalysis cycles, respectively.

3. The authors should fully characterize the conformation of the PEG chains in the COFs. It is still suspicious that the polymer chains would form an extended conformation along the 1-D channel direction. Note that PEG chain is very flexible and even could form globular structure in mesopores of COFs.

Response: Thanks for the insightful comment of the reviewer. To the best of our knowledge, PEG chains in the molten state resemble the unperturbed random coils. When the molten PEG chains are threaded into the pores, they are more likely to maintain the conformation of random coils instead of fully extended chains. The dimension of PEG coils in the molten state can be described by the radius of gyration. According to the earlier report (Ref. 40 Kawaguchi, S. et al. *Polymer*, 1997, 38, 2885-2891), the θ -condition of PEG 20k can give the radius of gyration of ~ 4.7 nm. Therefore, it is roughly assumed that the coil dimension of molten PEG chains is similar to this level, relatively larger than the pore size of BT-COF (2.4 nm). To minimize the entropy loss, the PEG coils have to be elongated to enter the 1D channels. Currently, experimentally characterizing the polymer coil dimensions in heterogeneous systems is technically challenging, which is typically conducted via the viscometric method in a diluted homogenous system.

The corresponding revision has been made in the context, *Page 17, Line 11-23*, “**To our knowledge, conformation of molten polymer chains resembles their unperturbed random coil in the θ -solvent. Thus, the PEG chains may still maintain random coils, instead of being completely extended, when threading the 1D pore channels of BT-COF. Such conformation ensures the minimum of entropy loss for the whole system. The coil dimension of molten PEG 20k is similar to the radius of gyration derived from the free PEG-20k coils in the θ -condition (~ 4.7 nm was obtained in 0.45 M K_2SO_4 aqueous solution at 35°C)⁴⁰. Compared with the pore size (2.4 m) and average crystallite size (44.6 nm) of BT-COF, it is conceivable that the molten PEG coils are elongated to adapt the split and hexagonal pore shape during the infiltration (Fig. 4a). When the accessible pore channels of BT-COF were stuffed with the deformed PEG coils, the interlayered dislocation was effectively suppressed to maintain the coaxial stacking. Indeed, with an increase in PEG amounts from 10wt.% to 30wt.%, the more densely filling of pore space made the ordering layered structure more stable, leading to the HERs rising from 8.90 mol g⁻¹ h⁻¹ to 11.14 mol g⁻¹ h⁻¹.”**

4. Considering the small domain size of COFs (Fig. 1d), the PEG chains are hardly encapsulated with a fully extended conformation in the COF channels, as was shown in Fig 4b.

Response: Thanks for the insightful comment of the reviewer. We agree that the PEG chains are hardly to attain a fully extended conformation along the 1D channels. We

then calculated the average crystallite size of BT-COF to compare with the coil dimension. The domain size of BT-COF, σ , could be determined by Scherrer equation, $\sigma = K\lambda/(\beta\cos\theta)$, wherein K is a shape factor (assumed to be 1), β is the full width at half maximum (FWHM), λ is the X-ray wavelength, and θ is the Bragg angle of the diffraction peak (Ref. 37. N. C. Flanders, et al., J. Am. Chem. Soc. 2020, 142, 14957). Based on the crystalline peak of the (100) plane, the FWHM was determined by fitting the peak with Lorentzian function, then remarking the value of β . The average crystallite size was calculated to be 44.6 nm (Supplementary Fig.18 and Table 2). This is large enough to accommodate the elongated PEG coils. According to the reviewer's comment, we have modified Fig. 4a to exactly illustrate the infiltration of long-chain PEG coils.

Fig. 4a Illustration of structural transformation of COFs and PEG@COF during the deposition of Pt cocatalyst.

5. The distance between the neighboring layers in the COF is 3.5 Å, as confirmed by PXRD patterns (Fig. 1b), which is comparable with the thickness of the PEG chains. Therefore, side diffusion of the polymer chains beyond the 1D channels could be also allowed. In this regard, again, the current data cannot support the threading of the polymer chains along the channel direction.

Response: Thanks for the issue addressed by the reviewer. We have carefully considered the possibility of side diffusion of the polymer chains, while it may hardly occur according to the experimental data and calculation results.

Strong interlayer interactions including π - π coupling and H-bonding ensure the coaxial stacking with ordered alignment in 2D COFs. Overcoming the strong interlayer interactions requires that PEG molecules should have much stronger interactions with COF skeletons. As computed for the binding of PEG with COF in energy (Section 4.3 in Supplementary Information), it is found that a short model of PEG has the larger binding energy at the side than on the surface with the cutout cluster of COFs. This

further signifies that long-chain polymers will likely bind to the sideways position stronger than those provided above. On the other hand, the interlayer distance of 3.5 Å requires that PEG chains intercalate COFs only via a fully extended conformation. The transformation of random coils to extended chains must lead to significant decrease in entropy, which is energetically unfavorable.

In control experiments, we compared the morphology and size of 30%PEG@BT-COF before and after ultra-sonication for 30 min in ethanol. The SEM images revealed that the treated COF aggregates were unchanged after sonication (**Supplementary Fig.17**), further ruling out the possibility that a number of PEG chains have diffused into the interlayer space between neighboring COF layers.

To clearly emphasize the point, we have modified the part in the context, *Page 9, Line 5-9*, **“On the other hand, it rules out the possibility that the PEG chains are embedded into the interlayered space, since the sandwiched PEG chains would have destroyed the ordered stacking and in turn, exfoliated the layered structures. Also, as proved by quantum chemistry calculations, this would not be energetically favored (see Section 4.3 in Supplementary Information for details).”**

6. The catalytic activity of Pt nanoparticles would be affected by their particle size and location (inside or outside the pores). In this regard, it should be better to provide the detailed characterizations of the Pt nanoparticle and investigate effects of PEG on the growth of Pt nanoparticles.

Response: Thanks for the constructive comment of the reviewer. As suggested, we investigated the effect of PEG on the growth of Pt nanoparticles. At first, the quantitative analysis was performed by ICP to confirm 3.54 wt.% Pt for BT-COF and 3.73 wt.% Pt for 30%PEG@BT-COF (**Supplementary Table 4**). Therefore, the incorporated PEG chains have the neglected effect on the amount of deposited Pt nanoparticles. Second, the size distribution of Pt nanoparticles in BT-COF and 30%PEG@BT-COF was rigorously analyzed. As shown in the TEM images (**Supplementary Fig. 30a-d**), the Pt nanoparticle are discretely distributed in the sample. Meanwhile, according to the statistical size distributions (**Supplementary Fig. 30e**), the photo-deposited Pt nanoparticles are populated at 1.9 nm in BT-COF and 2.3 nm in 30%PEG@BT-COF, respectively. Therefore, we concluded that the presence of PEG provides nucleation sites for the favorable growth of Pt nanoparticles and in turn, slightly broadens the size distribution. While HR TEM image shows that Pt nanoparticles are uniformly distributed on the framework of BT-COF and 30%PEG@BT-COF; therefore, the position distribution of nanoparticles has an equivalent effect on the catalytic performance of both COFs.

The supplemented data has been provided in **Supplementary Fig. 30** and the revision has been made in the context, *Page 14, Line 16-22*, and *Page 15, Line 1-2*, **“HR TEM images displayed a uniform deposition of Pt nanoparticles, while a close inspection at the size distribution revealed that the 30%PEG@BT-COF resulted in the**

dominated size of 2.3 nm, which was a little larger than that (1.9 nm) of Pt nanoparticles produced within the BT-COF (Supplementary Fig. 30). This is probably due to the infiltrated PEG that provides a number of nucleation sites for the favourable growth of Pt nanoparticles⁴⁰ and subsequently the broadened size distributions. As the increasing sizes of Pt nanoparticles usually lower the catalytic activity⁴¹, the enhanced photocatalytic performances obtained from 30%PEG@BT-COF should be marginally affected by the slight change of Pt nanoparticles on sizes.”

Fig. 30 (a-d) TEM images of the Pt nanoparticles photo-deposited on BT-COF (a,c) and 30%PEG@BT-COF (b,d), respectively. (e,f) The statistical size distributions of Pt nanoparticles in BT-COF (e) and 30%PEG@BT-COF (f), respectively.

7. What is the peak at 187 ppm in ¹³C CP-MAS NMR spectrum of BT-COF (Fig. S2)?

Response: Thanks for the reviewer pointing out the missing peak. As reported in the earlier studies (S. Ghosh, et al. *J. Am. Chem. Soc.* **2020**, *142*, 9752–9762), the keto form of carbon atoms often exhibits the multiple chemical shifts from ~180 ppm to 188 ppm. Therefore, we still assigned the peak at 187 ppm to carbonyl carbon of BT-COF in the ¹³C CP-MAS NMR spectrum. This has been mentioned in the revised context, *Page 5, Line 6-9*, “Solid-state ¹³C CP/MAS NMR spectrum verified the presence of carbonyl C(a) at 185–187 ppm and secondary amine C(b) at 147 ppm, respectively, as well as the other aromatic signals corresponding to the phenyl C at 114–138 ppm and benzothiadiazole C(d) at 153 ppm (Supplementary Fig. 2).” And the peak is marked in the ¹³C CP-MAS NMR spectrum, *Supplementary Fig. 2*.

Supplementary Fig. 2 Solid-state ^{13}C CP-MAS NMR spectrum of BT-COF.

8. Bo Wang et al. has already reported hybridization of PEG and COF (Bo Wang, et al., J. Am. Chem. Soc., 141, 1923). This work is highly relevant to the present work and should be cited.

Response: According to the reviewer, we have cited this paper in the main text, *Page 7 Ref. 32*, “**32. Guo, Z. et al. Fast ion transport pathway provided by polyethylene glycol confined covalent organic frameworks. J. Am. Chem. Soc. 141, 1923–1927 (2019).**”

9. Stacking structure of the COFs might be retained after the introduction of the PEG chains into the COFs, as confirmed by the wide-angle X-ray scattering measurement (WAXS) measurements (Fig. 2). However, I am not still convinced that the crystalline structures of the COFs were completely maintained, as shown in Fig. 4b. There is no direct evidence for the ordering along the (110) and (210) planes.

Response: Thanks for the insightful comment. For clearly showing the other crystalline peaks, the electron density contrast was further enhanced in the 2D X-ray scattering image, the three new peaks emerged in the WAXS patterns of 30%PEG@BT-COF, which could be assigned to the (110), (200), and (210) lattice planes, respectively. This clearly evidences that the crystalline characteristic of PEG@BT-COF is maintained. The observation further proves that the PEG chains would not diffuse into the interlayers of BT-COF to destroy the ordering stacks.

The data in Fig. 2b is updated and the discussion is added in the main text, *Page 8, Line 16-18*, “**As displayed in Fig. 2b, the dominated peaks at 2.79° corresponding to the**

(100) facets can be clearly detected in 30%PEG@BT-COF, as well as the other refined peaks ascribed to the (110), (200) and (210) planes.”

Fig. 2b WAXS profile of 30%PEG@BT-COF. Inset is the enlarged region to observe the peaks for the (110), (200) and (210) lattice facets.

10. The authors should perform quantitative analysis for the degree of structural maintenance upon the insertion of the polymer chains. Is it possible to discuss the crystallinity using WAXS?

Response: We greatly appreciate the valuable suggestion of the reviewer. Hereby, we performed the quantitative analysis for the degree of structural maintenance upon the infiltration of PEG. By using the Scherrer equation, the crystalline domain sizes were computed via the preferential (100) crystal facet, which can remark the average crystalline domains of BT-COF (A. L. Patterson, *Phys. Rev. B.* 1939, 56, 978–982; *Nanoscale science and technology.* 2005, John Wiley, Chichester, England.). From the WAXS pattern, the parent BT-COF gives the average crystallite size of 44.6 nm. After infiltration of PEG chains into the BT-COF, the calculated domain size reduced to 38.4 nm, reflecting that the crystalline domain of BT-COF was slightly reduced. As the keto-enamine linkages are relatively stable, it is not likely that PEG filling causes the bond cleavage and structural collapse. We thus reason that the dynamic change of the flexible COF skeletons in conformation may occur to adapt the PEG filling (D. Zhao, et al. *J. Am. Chem. Soc.* 2020, 142, 30, 12995–13002). After 48-h photocatalytic cycles, the crystalline domain sizes of the recycled 30%PEG@BT-COF was calculated to be 33.1 nm, maintaining 86.1% of the crystalline size compared with that before photocatalysis. In contrast, BT-COF without PEG stabilization lost 51.6% of crystalline domains after 48-h photocatalysis, remaining the average crystallite size of 21.6 nm. All the findings well corroborated that the incorporated PEG played the critical role on the ordering structural maintenance.

The structural maintenance has been analyzed in the revised context, *Page 9, Line 20-23* and *Page 5, Line 1-5*, “**Then a quantitative analysis of structural durability was performed by calculating the average crystallite sizes with Scherrer equation, $\sigma =$**

$K\lambda/(\beta\cos\theta)$, wherein K is a shape factor (assumed to be 1), β is the full width at half maximum (FWHM), λ is the X-ray wavelength, and θ is the Bragg angle of the diffraction peak. FWHM was obtained by fitting the strongest (100) X-ray scattering peak in the WAXS patterns (Supplementary Fig. 18 and Table 2)³⁷. Compared with the parent BT-COF (44.6 nm), the crystalline domain size of 30%PEG@BT-COF was reduced to be 38.4 nm, preserving 86.1% of the initial size. Therefore, it is conclusively validated that filling the mesopores with long PEG chains can preserve a majority of the layered ordering stack for the BT-COF.”, and *Page 16, Line 17-21*, “By using Scherrer equation, we found that the average crystallite sizes of PEG@BT-COF slightly decreased from 38.4 nm to 33.1 nm before and after the photocatalytic cycles, maintaining 86.2% of the initial domain size (Supplementary Fig. 18 and Table 2). In sharp contrast, there was merely a half of crystallite size (48.4%) remaining in the recycled BT-COF (21.6 nm) under the identical conditions of photocatalytic cycling.” The supplemented data was added in Supplementary Fig. 18 and Table 2.

Supplementary Table 2. FWHM (R^2) and average crystallite domain sizes computed by Scherrer equation.

	FWHM (R^2) ^a	Average crystallite domain sizes (nm)
BT-COF	0.1981 (0.9910)	44.6
Recycled BT-COF	0.4090 (0.9977)	21.6
30%PEG@BT-COF (PEG = 20k)	0.2299 (0.9890)	38.4
Post-extracted PEG@BT-COF ^b	0.2784 (0.9778)	31.8 ^b
Re-30%PEG@BT-COF	0.2670 (0.9935)	33.1
Post-extracted re-30%PEG@BT-COF ^b	0.2997 (0.9810)	29.6 ^b

^aFWHM(R^2) values were obtained from Supplementary Fig. 18.

^bThe samples were extracted by THF to remove the loaded PEG chains. As compared with those before solvent post-extraction, the average crystallite sizes decrease a little. This is possibly resulted from the solvation exfoliation effect.

Supplementary Fig. 18 The (100) X-ray scattering peaks fitted by Lorentzian function for obtaining the full width at half maximum (FWHM).

Reviewer #3:

The authors describe the structural stabilization of 2D COFs by incorporating PEG chains into the pores. The authors claim that this threading of polymer chains into the pores is stabilizing the ordered arrangement of the stacking layers, especially under photocatalytic conditions. Indeed the PEG infiltrated COFs show higher and more stable photocatalytic hydrogen evolution from water. The COF and COF-PEG composites are analyzed with a variety of methods to explain the effect of the PEG infiltration.

I think this is an interesting addition to the fast growing field of photocatalysis with COFs, even though the improvement by adding PEGs is not such pronounced. As the authors show, small variations in Pt content or pH already have a more distinct effect on the hydrogen evolution yield than the PEG threading.

Anyway, because of the high current interest in COFs and photocatalytic HER, the work can be suitable for Nature Communication.

There are just some minor concerns:

1. In general, the authors have chosen to combine many schemes and measurements in one figure. This results in very small diagrams. Just as example all XRD patterns in Figure 1 g can be hardly recognized, and if the “crystalline structure is retained without any signs of destruction” is thus hard to judge. Some measurements should be moved to the SI or, if possible, shown their again in higher resolution.

Response: Thanks for the suggestion of the reviewer. We have revised the layout of **Fig. 1-4**, move Fig. 2a to Supplementary Information (**Supplementary Fig. 2**), and increase the size and resolution of **Fig. 1f and 1g**.

2. For the stability tests, not just the XRD pattern before and after treatment should be compared but also the sample weight. If some amount of the COF is getting decomposed and dissolved, this is not necessarily seen in the PXRD of the residue.

Response: Thanks for the valuable suggestion of the reviewer. We further evaluated the weight loss of the BT-COF after treatment with different solvents. As shown in **Supplementary Fig. 8**, the treated BT-COF samples show approximately < 3 wt% of weight loss and no residue in these solutions, verifying the chemical stability of BT-COF in these solvents.

The revision has been made in the main text, *Page 7, Line 1-2*, “**The weight loss of the treated COFs was nearly ignorable and the solutions appeared colourless, indicating no decomposition inspected in such treatment (Supplementary Fig. 8).**”

Supplementary Fig. 8 The remaining weights (a) and supernatants photographs (b) for the treated BT-COF with different solvents including THF, DMF, HCl (12 M) and NaOH aq. solution (1M), respectively.

3. That the “thermogravimetric analysis proved again that 30wt% PEG or less was loaded in BT-COF” is a quite unspecific statement, especially as values are given in the figure caption of Figure S8. However, as there is no clear step for PEG decomposition in the TGA, these loading values should be handled with care, especially to give numbers with one digit seem not trustworthy. Which “theoretical values” are meant in the figure caption of Figure S8?

Response: Thanks for the reviewer pointing out the unclear parts. We offered the accurate values of the loaded PEG according to the TGA results. The changes have been made on **Supplementary Fig. 9**, remarking the PEG weight loss. The term “theoretical values” refers to the relative ratio of the feeding PEG to BT-COF.

The corresponding revision has been made in the main text, *Page 7, Line 21-23 and Page 8, Line 1-2*, “Also, the thermogravimetric analysis showed that the thermal decomposition of PEG chains appeared in the range from 350°C to 450°C, giving a corresponding weight loss of 9.1%, 16.6% and 22.7%, respectively, which matches well with the filling of 10%, 20% and 30% PEG in the BT-COFs (Supplementary Fig. 9).”

Supplementary Fig. 9 TGA profiles of 10%PEG@BT-COF (a), 20%PEG@BT-COF (b) and 30%PEG@BT-COF (c), respectively. As the temperature range for the PEG thermal decomposition is determined in the range from 350°C to 450°C, the weight percentages of the loaded PEG relative to the total mass of PEG@BT-COF are found at 9.1%, 16.6% and 22.7%, respectively. They are almost the same with the feeding contents, i.e. 9.09%, 16.67% and 23.08%, calculated from the 10%PEG@BT-COF, 20%PEG@BT-COF and 30%PEG@BT-COF, respectively.

4. The authors should describe the difference between their PXRD and WAXS measurements. Especially as the “WAXS” measurement show also very small angles, thus this seem not to be a conventional WAXS.

Response: Thanks for the kind suggestion of the reviewer. We performed the WAXS measurements on the XEUSS SAXS/WAXS system, which allows a broad scanning angles ranged from 0.035 to 60 degree. The primary difference between PXRD and WAXS is summarized in the following table. For the PXRD characterization, disorder at the molecular level leads to the peak broadening in the diffraction pattern, and can eventually result in the disappearance of the diffraction peaks with increasing disorders. In the case, the existing crystalline domains cannot be precisely identified in composites by PXRD. In contrast, the WAXS technique can reveal the electron density structures applicable to the crystalline, amorphous and semi-crystalline materials. Considering that the variation of crystalline domains might not be precisely revealed by the PXRD

characterization due to the presence of non-crystalline PEG chains, the WAXS characterization is used to characterize the atomic ordering of BT-COF component in the PEG@BT-COF by distinguishing electron density contrast.

	PXRD	WAXS
Pressure	ambient	High vacuum
Voltage	40 kV	50 kV
Current	40 mA	0.6 mA
Angular range	2-60 degree	0.035-60 degree

The detailed information regarding the WAXS measurement has been given in *Supplementary Information, Page S3*, “**Wide Angle X-ray Scattering (WAXS) measurements were performed on the Xeuss 2.0 with a Pilatus 3R 200K-A detector. Copper K α ($\lambda = 1.54 \text{ \AA}$) was used as the radiation source. The distance between the sample and the detector is 148.35 mm. The exposing time is 600 s. The scanning degree is ranged from 0.035 to 60 degree.**”

5. In both solid state NMR measurements there is an obvious peak at 25 ppm (plus some smaller ones at this area), which are certainly no spinning side bands. How can these peaks assigned?

Response: We appreciate that the reviewer pointed out the missing assignment in the NMR spectrum. The chemical shift at 25 ppm was ascribed to alkane carbon atoms originated from the terminal groups containing the tertiary amines when pyrrolidine was used as a catalyst. This has been reported in our earlier work, **Ref. 28**, “**28. Wang, R., Kong, W., Zhou, T., Wang, C. & Guo, J. Organobase modulated synthesis of high-quality b-ketoenamine-linked covalent organic frameworks. Chem. Commun. 57, 331-334 (2021).**” Meanwhile, the presence of alkane carbons was also verified by the characteristic vibrations of alkyl C-H appeared at 2857 cm^{-1} in FT IR spectra (Figure R1).

The corresponding revision has been made in the main text, *Page 5, Line 7-9*, “**The chemical shift at 25 ppm was ascribed to alkane C(f) originated from the terminal groups containing the tertiary amines arising from the reaction between Py and aldehydes²⁸.**”

Supplementary Fig. 2 Solid-state ^{13}C CP-MAS NMR spectrum of BT-COF.

Figure R1. FT IR spectrum of BT-COF synthesized using pyrrolidine as a catalyst.

6. The increasing stability of layer stacking is sacrificed by an overall lower surface area of the COF. Thus is the apparent surface area of a COF photocatalyst not such important as it is claimed in recent literature?

Response: Thanks for the insightful comment of the reviewer. The photocatalytic reactions involve in several sequential steps after light absorption, including charge separation and migration, surface reaction and substance adsorption and desorption. Therefore, many different factors can affect the photocatalytic performances such as band structures, surface active sites and charge separation (*e.g.*, K. Takane, ACS

Catal. 2017, 7, 8006-8022; A. Kudo and Y. Misekita, Chem. Soc. Rev. 2009, 38, 253-278.). The final photocatalytic performance is the outcome of the combined chemical and physical properties of the photocatalysts.

In this work, the band structure of BT-COF and 30%PEG@BT-COF is relatively similar (Fig. 3b), indicating that the two COFs possess the similar thermodynamic driving force and can absorb the similar amount of incident light for photocatalysis (Fig. 3a). The 30%PEG@BT-COF shows a smaller specific surface area compared to BT-COF but exhibits much higher photocatalytic performance. The reason may lie in that the active sites for photocatalytic proton reduction are the deposited Pt nanoparticles but not the carbon/nitrogen atoms in the COFs. Therefore, the inherent surface area of COFs should not affect the photocatalytic activity of COFs. Furthermore, the results from transient photocurrent, fluorescence decay profiles, and Nyquist plots measurements reveal that the charge separation is more efficient in 30%PEG@BT-COFs compare to BT-COF. Overall, these results suggested that incorporating PEG chains into the BT-COF can not only promote charge separation but also enhance stability of the BT-COF.

7. Even though the absolute values are not such high, the effect of PEG threading on HER seem to be more pronounced for TP-COF than for BT-COF. Can this be due to an increased hydrophilicity of TP-COF when PEG chains are introduced? At least in water sorption measurement TP-COF seem to have low water uptake, a water sorption measurement of PEG@TP-COF would be therefore interesting.

Response: Thanks for the valuable suggestion of the reviewer. We conducted the water sorption measurement for the 30%PEG@TP-COF. As shown in **Supplementary Fig. 23**, the water uptake of the PEG@TP-COF decreased from 167 to 122 cm³ g⁻¹ compared with the TP-COF. However, the decreased tendency of water uptake was more remarkable for the BT-COF (from 643 to 244 cm³ g⁻¹) compared to the TP-COF when loading the same amount of PEG. The results mean that the loss of water uptake capacity is smaller for 30%PEG@TP-COF. Thus, the performance improvement is much more obvious compared with 30%PEG@BT-COF.

The revision has accordingly been made with the added data, **Supplementary Fig. 23, Page 10, Line 22-23 and Page 11, Line 1-4**, “**As the mesopores of BT-COF and TP-COF were filled up with high-molecular-weight PEG chains, the total water uptake both decreased and the tendency was more remarkable for BT-COF than TP-COF when loading the same amount of PEG. As a result, the loss of water uptake capacity is smaller for 30%PEG@TP-COF (Fig. 2e and Supplementary Fig. 23), thereby which is more beneficial for water splitting in photocatalysis.**”

Supplementary Fig. 23 Water uptake profiles of TP-COF and 30%PEG@TP-COF at 298 K, respectively.

REVIEWER COMMENTS

Reviewer #1 (Remarks to the Author):

I have carefully gone through all the reports of the reviewers and the author's response. Authors have answered all the questions satisfactorily and they have also addressed the concerns of the 2nd reviewers critically. I am recommending the manuscript for publication.

Pranab Sarkar

Reviewer #2 (Remarks to the Author):

In the revised version, the authors tried to address the concerns from reviewers and could improve the manuscript decently. I felt the revision is still insufficient for the acceptance for publication; moreover, this revision raises additional questions.

1) Although the authors claimed this is the first study that could improve the stability of COF by polymer insertion, the same concept for the structural stabilization of framework materials, such as MOFs, have been already reported in many papers, as pointed out in the previous round of review. In their rebuttal, they described “Whether the polymer-infiltration strategy is effective in stabilizing porous materials towards a catalytic reaction is still not known. We believe that our study provides significant insights into the structural stability of COFs during photocatalysis and offers new perspectives in developing porous materials for a wide range of catalytic applications via this polymer-infiltration strategy.” This is very narrow scope that should be involved in the general concept of “structural stabilization of framework materials via polymer insertion”. The authors should reconstruct the introduction part to cite them and appropriately describe the development of this field.

2) Considering the radius of gyration of PEG20K, the PEG chains cannot be “extended” or “elongated” along the COF channels with 2.4 nm diameter. Use of these words would mislead the readers. The authors should carefully determine the conformation of PEG accommodated in COF.

3) The size of Pt nanoparticles formed in the PEG-accommodated COF is larger than the pore of COF, suggesting the formation of Pt nanoparticles outside the COF channels. I am concerned about the possibility that the Pt particles outside the COF would substantially enhance the photocatalytic performance.

Reviewer #3 (Remarks to the Author):

The authors have adequately answered all concerns of this reviewer and therefore I can recommend this work for publication.

As additional note: I'm slightly surprised by the relatively large amount of terminal groups in the COFs (see answer to comment #5). These cannot be quantified of course from CP-MAS NMR, but at least the peaks are clearly visible, which is not often seen for other COFs (or probably just not mentioned as in the first version of this work). The authors state that these peaks have been reported in an earlier work by themselves. I wonder if also other groups have seen pronounced peaks for terminal groups when using pyrrolidone or other catalysts?

Regarding the answer to comment #7. That the total water uptake capacity decreases after loading PEG into the pores of the COFs is not totally surprising. An increased hydrophilicity is however not seen necessarily by a change in total uptake but by the shape of the isotherm, especially water uptake at low pressures. Comparing the isotherms indeed a slightly higher uptake of water at lower pressures is seen for PEG@TP-COF, but probably this is not such pronounced that it must be further discussed or highlighted. I just think that the discussion of change in total water uptake is also not conducive when discussing the increased photocatalytic performance.

REVIEWER COMMENTS

Reviewer #1:

I have carefully gone through all the reports of the reviewers and the author's response. Authors have answered all the questions satisfactorily and they have also addressed the concerns of the 2nd reviewers critically. I am recommending the manuscript for publication.

Pranab Sarkar

Response: We greatly appreciate the positive recommendation from the reviewer.

Reviewer #2:

In the revised version, the authors tried to address the concerns from reviewers and could improve the manuscript decently. I felt the revision is still insufficient for the acceptance for publication; moreover, this revision raises additional questions.

1) Although the authors claimed this is the first study that could improve the stability of COF by polymer insertion, the same concept for the structural stabilization of framework materials, such as MOFs, have been already reported in many papers, as pointed out in the previous round of review. In their rebuttal, they described “Whether the polymer-infiltration strategy is effective in stabilizing porous materials towards a catalytic reaction is still not known. We believe that our study provides significant insights into the structural stability of COFs during photocatalysis and offers new perspectives in developing porous materials for a wide range of catalytic applications via this polymer-infiltration strategy.” This is very narrow scope that should be involved in the general concept of “structural stabilization of framework materials via polymer insertion”. The authors should reconstruct the introduction part to cite them and appropriately describe the development of this field.

Response: Thanks for the constructive comment from the reviewer. Indeed, the insertion of polymer chains into MOFs has been extensively investigated. Those studies should be properly cited and discussed in the manuscript. Now we have revised our introduction to include the previous studies (see below).

Page 4, Line 1-9, “Entrapping functional guests into porous structures has been demonstrated to be a flexible and effective strategy to modify the pore environment, most notably with linear polymers. In the pioneering studies, the polymer-threaded metal-organic frameworks have well preserved structural stability and maintained accessible surface areas and opening pores, albeit with exposure to pressure²⁷, heat²⁸ or solvent treatment²⁹. Meanwhile, the purposeful incorporation of functional polymers into pore channels can broaden functionality and enhance performances of framework materials for a wide range of applications³⁰⁻³⁵. These inspiring studies motivate us to explore a polymer-infiltrated 2D COF with the aim to develop an

innovative strategy for boosting the photocatalytic H₂ evolution in water splitting.”

The added references are shown below.

27. Celeste, A. et al. Mesoporous metal–organic framework MIL-101 at high pressure. *J. Am. Chem. Soc.* 142, 15012–15019 (2020).

28. Ouay, B. L., Takaya, H. & Uemura, T. Controlling the packing of metal–organic layers by inclusion of polymer guests. *J. Am. Chem. Soc.* 141, 14549–14553 (2019).

29. Peng, L. et al. Preserving porosity of mesoporous metal–organic frameworks through the introduction of polymer guests. *J. Am. Chem. Soc.* 142, 12397–12405 (2019).

31. Kitao, T., Zhang, Y., Kitagawa, S., Wang, B. & Uemura, T. Hybridization of MOFs and polymers. *Chem. Soc. Rev.* 46, 3108–3133 (2017).

32. Uemura, T. et al. Highly photoconducting π -stacked polymer accommodated in coordination nanochannels. *J. Am. Chem. Soc.* 134, 8360–8363 (2012).

33. Ouay, B. L. et al. Nanostructuring of PEDOT in porous coordination polymers for tunable porosity and conductivity. *J. Am. Chem. Soc.* 138, 10088–10091 (2016).

2) Considering the radius of gyration of PEG20K, the PEG chains cannot be “extended” or “elongated” along the COF channels with 2.4 nm diameter. Use of these words would mislead the readers. The authors should carefully determine the conformation of PEG accommodated in COF.

Response: Thanks for the kind reminder.

As suggested by the reviewer, we performed the coarse-graining simulation in a canonical ensemble to study the conformation of PEG chains in the pore channel. With the dissipative particle dynamics (DPD) method, a PEG chain was modelled by the linked DPD beads, which had the similar radius of gyration with a PEG-20k coil (~5.0 nm) in the θ -condition. When such a model was threaded in a given nanotube ($r = 1.2$ nm), it had to be elongated to adapt the confined space. Meanwhile, some segments of the PEG chains would adhere to the pore wall due to the hydrogen bonding between the threaded PEG chain and the pore wall. On basis of the simulated results, it can be concluded that the term “elongated” is appropriate for describing the conformation of PEG coils in the pores. In fact, this is not a special case. The similar theoretical studies have been reported in the earlier publications (e.g. **Macromolecules**, 2013, 46, 6336). Also, we note that some reports claimed that the conformation of infiltrated polymers was “extended” in the MOFs (e.g. **J. Am. Chem. Soc.** 2015, 137, 5231; **Chem. Sci.**, 2020, 11, 10844).

The revision has been made in the main text, **Fig 4a** is correspondingly edited according to the simulation results. The changes are shown in **Page 17, Line 21-23** and **Page 18, Line 1-18**, “After the linear PEG chains are incorporated into the pore channels, the random coils are reasonably stretched in the constrained space, although a fully extended conformation is not thermodynamically allowed as the minimum of entropy loss must be ensured during the conformation transition. To confirm our speculation on the change of the PEG conformation, a dissipative particle dynamics (DPD) simulation was performed to investigate the universal

natures of polymer chains in a coarse-graining route. PEG was modelled by DPD beads sequentially linked by harmonic bond and the water solvent was modelled as single beads. The hard repulsive Lennard-Jones potential was applied to implement the constraint of the nanotube to the PEG and solvent beads. Moreover, the hydrogen bonding was added between the PEG-beads and the nanotube wall. As displayed in Fig. 4a, the radius of gyration ($R_g = 5.0$ nm) for the random coil in the bulk estimated by DPD simulations is similar to that of PEG-20k in the θ -condition⁴⁶. By a quantitative comparison, R_g of the PEG-20k model in the bulk is approximately 4 times the radius of the nanotube. Consequently, the PEG-chains in the nanotube have to be elongated and partially adhered to the pore walls via the hydrogen bonding. When the radius of the nanotube is expanded, the stressed PEG chain is relaxed to some extent but still retains the elongated state (Supplementary Fig. 42). Therefore, once a number of pore channels of BT-COF were threaded with the elongated PEG coils, the interlayered dislocation can be effectively suppressed to maintain the coaxial stacking.”

Fig. 4a Illustration of structural transformation of COF and PEG@COF during the deposition of Pt cocatalyst. PEG conformations in water and in a pore channel (2.4 nm) are simulated by the dissipative particle dynamics (DPD) method in the canonical ensemble, respectively. According to the coarse-graining model, harmonic-bond-linked DPD beads and individual beads are applied to simulate PEG and water, respectively.

Supplementary Fig. 42 (a) DPD simulation for the conformations of PEG chains (the linked 80 red beads) and water solvents (single blue beads) in a variety of nanotubes with the radius of 1.2 (I), 1.6 (II), 2.0 (III), 2.4 (IV), 2.8 (V), 3.2 (VI), and 3.6 (VII), respectively. (b) The distance r vs. Lennard-Jones repulsion interaction between nanotube wall and PEG-chain model. (c) The distance r vs. H-bonding attraction interaction between nanotube wall and PEG-chain model. (d) The radius of gyration vs. the radius of the nanotube. The fitted curves represent the cylinder constraint state, the parallel straight lines represent the bulk solvent state, and N represents the number of repeated beads in the polymeric chain.

The detailed simulation method has been added in the Supplementary Information, Page S13-S17.

4.6 Theoretical simulation of polymer conformation.

Simulations were performed by using dissipative particles dynamics (DPD) method in the canonical ensemble, which is a coarse-gained particle-based mesoscopic simulation technique developed by Hoogerbrugge and Koelman in 1992.^[S24] The coarse-graining approach is able to capture universal properties of polymers rather than the interactions between specific chemical functional groups. The time evolution of DPD beads with unit mass is governed by Newton's equations of motion:^[S25]

$$\frac{dr_i}{dt} = v_i, \quad \frac{dv_i}{dt} = f_i$$

$$f_i = \sum_{j \neq i} (F_{ij}^C + F_{ij}^D + F_{ij}^R),$$

in which the force is composed of conservative force F_{ij}^C , dissipative force F_{ij}^D , and random force F_{ij}^R . All forces are pairwise-additive, repulsive and short-range with a cutoff at $r_c = 1$.

The conservative force F_{ij}^C is a soft-repulsive interaction acting along the line of the centers of two particles:

$$F_{ij}^C = \begin{cases} a_{ij}(1 - r_{ij})\hat{r}_{ij} & (r_{ij} < 1) \\ 0 & (r_{ij} \geq 1) \end{cases},$$

where r_{ij} denotes the distance between beads i and j ,

$$r_{ij} = r_i - r_j,$$

$$r_{ij} = |r_{ij}|,$$

and \hat{r}_{ij} represents the unit vector pointing from j to i ,

$$\hat{r}_{ij} = r_{ij}/|r_{ij}|.$$

The interaction parameter a_{ij} can be estimated from the Flory–Huggins χ -parameter by $a_{ij} \approx 25 + 3.27\chi_{ij}$.^[S25] a_{ij} was chosen to be 30 between PEG beads, and $a_{ij} = 25$ between the solvent and PEG beads to mimic the solubility of PEG in water.

The dissipative force F_{ij}^D is proportional to the relative velocity and takes the form:

$$F_{ij}^D = -\gamma w^D(r_{ij})(\hat{r}_{ij} \cdot v_{ij})\hat{r}_{ij},$$

where $v_{ij} = v_i - v_j$ is the relative velocity and γ is the friction coefficient.

The random force F_{ij}^R acts as a heat source to equilibrate the thermal motion and takes the form:

$$F_{ij}^R = \sigma w^R(r_{ij})\theta_{ij}\hat{r}_{ij}$$

Where σ specifies the noise strength and is set to be 3. $\theta_{ij}(t)$ is a randomly fluctuating variable with zero mean and unit variance, satisfy Gaussian statistics:

$$\langle \theta_{ij}(t) \rangle = 0,$$

$$\langle \theta_{ij}(t)\theta_{kl}(t') \rangle = (\delta_{ik}\delta_{jl} + \delta_{il}\delta_{jk})\delta(t - t').$$

The random force is related to the dissipative force so that they satisfy the fluctuation–dissipation relation:

$$\sigma^2 = 2\gamma k_B T$$

$$w^D(r) = [w^R(r)]^2 = \begin{cases} (1 - r)^2 & (r < 1) \\ 0 & (r \geq 1) \end{cases}$$

where w^D and w^R are the r -dependent weight functions vanishing at $r > r_C = 1$. Both dissipative force and random force act along the line of centers so that the linear and angular momentums are conservative, ensuring that the simulation is performed in a canonical ensemble.

The beads on the polymeric chain are connected by a harmonic spring potential as:

$$V_{\text{bond}}(r) = \frac{1}{2}k_b(r_b - r_0)^2$$

Where r_0 is the equilibrium length and was set to be 0.75, r_b is the distance between connected beads, k_b is the spring constant and was chosen to be 30.0.

The PEG chain was modeled by DPD beads sequentially linked by harmonic bond, and it was confined in a nanotube, while the water solvent was modeled by single-bead molecules.

The constraint of the nanotube to the PEG and solvents was implemented by the hard-repulsive Lennard-Jones potential:

$$V_{LJ}(r) = \begin{cases} 4\varepsilon \left[\left(\frac{\sigma}{r}\right)^6 - \left(\frac{\sigma}{r}\right)^{12} \right] & r < r_{cut} \\ 0 & r \geq r_{cut} \end{cases}$$

Where ε is the depth of the potential at its minimum and was set to be 1.0, r is the distance between PEG/solvent beads and the pore wall, and $r_{cut} = 1.1225$ is the cutoff of the Lennard-Jones force.

The hydrogen bonding attractive interaction between PEG and nanotube was qualitatively mimicked by a Morse interaction as:

$$V_{morse}(r) = \begin{cases} D_0 \left[\exp(-2\alpha(r - r_0)) - 2\exp(-\alpha(r - r_0)) \right] & r < r_{cut} \\ 0 & r \geq r_{cut} \end{cases}$$

Where D_0 is the depth of the potential at its minimum and was set to be 1.0, α controls the width of the potential well and was set to be 3.0, r_0 is the position of the minimum and was set to be 1.0, and r_{cut} is the cutoff of the Morse force and was set to be 3.0.

The number density of beads was fixed as 3 by setting total number of polymeric and solvent beads as well as the sizes of the simulation box. The system temperature was maintained at $T^* = 1.0$. The radius of interaction between molecules and the particle mass were set to unity, leading to the reduced unit of dimensionless time of the system as $\tau = R_c \sqrt{m/k_B T}$. The Velocity-Verlet integration scheme was used to integrate the equations of motion. The time-step was set to be 0.001τ to achieve a balance between simulation stability and performance.

All the simulated systems were randomly positioned in the simulation box and were simulated $1 \times 10^3 \tau$ to generate a disordered state. Simulations of $1 \times 10^4 \tau$ were performed to anneal the system toward the equilibrium morphology.

The initial configuration was generated by the open-source package GALAMOST (version 4.0.1), developed and maintained by Zhu.^[S26] Simulations were performed using HOOMD-Blue,^[S27] v.2.9.3, a free and open-source code developed and maintained at the University of Michigan.

The modeled polymer chain was placed in a cylindrical tube with periodic condition in the z direction and no space constraint solvent state, respectively. The radius of the tube and the number of the beads in the polymeric chain was varied to investigate the confinement effect of the nanotube on the conformation of the PEG chain. As a comparison, the conformation of a single PEG chain in the bulk solvent is also studied in our simulations. Specifically, we calculate the mean square radius of gyration of the PEG chain confined in nanotubes with various diameters as well as that of the PEG chain in the bulk solvent. The mean square radius

of gyration is calculated by the following expression:

$$\langle R_g^2 \rangle = \frac{1}{N^2} \sum_{i=1}^N \sum_{j=1}^N \langle (\vec{R}_i - \vec{R}_j)^2 \rangle$$

Where N is the number of beads in the PEG chain, \vec{R}_i and \vec{R}_j is the position vector of bead i and j . For every point in Supplementary Fig. 42d, 200 samples were sampled to calculate the mean of the radius of gyration. The elongated conformation of the PEG chain in small nanotubes is clearly evidenced by our simulation results.

[S24] P. J. Hoogerbrugge, J. M. V. A. Koelman, Simulating Microscopic Hydrodynamic Phenomena with Dissipative Particle Dynamics. *Europhys. Lett.* 19, 155-160 (1992).

[S25] R. D. Groot, P. B. Warren, Dissipative particle dynamics: Bridging the gap between atomistic and mesoscopic simulation. *J. Chem. Phys.* 107, 4423-4435 (1997).

[S26] Y.-L. Zhu et al., GALAMOST: GPU-accelerated large-scale molecular simulation toolkit. *J. Comput. Chem.* 34, 2197-2211 (2013).

[S27] C. L. Phillips, J. A. Anderson, and S. C. Glotzer, Pseudo-random number generation for Brownian Dynamics and Dissipative Particle Dynamics simulations on GPU devices. *J. Comput. Phys.* 230, 7191-7201 (2011).

3) The size of Pt nanoparticles formed in the PEG-accommodated COF is larger than the pore of COF, suggesting the formation of Pt nanoparticles outside the COF channels. I am concerned about the possibility that the Pt particles outside the COF would substantially enhance the photocatalytic performance.

Response: Thanks for the insightful comment.

We agree with the reviewer that the Pt nanoparticles deposited outside the 30%PEG@BT-COF could be a factor to enhance the photocatalytic performance, but through the analysis and comparison of all results, we believe that this factor cannot be a crucial one.

(1) The photoexcited COFs provide photo-generated electrons for the reduction of Pt⁴⁺ ions into Pt nanoparticles. It is thus reasonable that the formed Pt nanoparticles are randomly distributed either inside or outside the COFs. During the photocatalytic process, water molecules can freely diffuse throughout the COF materials. Accordingly, the water reduction reaction occurs simultaneously on the Pt nanoparticles inside and outside the COF pores.

(2) Furthermore, we evaluated the effect of the infiltrated short PEG ($M_w = 2000$) chains on the formed Pt particle size and distribution. With the same photo-deposition method, it is found that the average size of Pt nanoparticles is around 2.2 nm, which is similar to that (~2.3 nm) obtained on the PEG-20k@BT-COF. The result implies that the short PEG chains can also promote the growth of Pt nanoparticles, leading to their preferable distribution outside the pores of BT-COF. Nevertheless, the HER of PEG-20k@BT-COF is *ca.* 1.4-fold higher than that of PEG-2000@BT-COF. This validates that the PEG-stabilized COF structures play an important role in the photocatalytic water splitting instead of the increasing number of Pt nanoparticles

outside the COFs.

(3) On the other hand, increasing the amount of H_2PtCl_6 could produce more and larger Pt nanoparticles on the surface of BT-COF. However, this attempt failed to further improve the photocatalytic performance. As shown in Supplementary Fig. 29, the HER for the BT-COF containing 9wt% Pt is lower than those for the 5wt%- and 7wt%-Pt loaded COFs. The possible reason lies in that the increase in sizes of Pt nanoparticles can impair the photogenerated electron transfer (ACS Catal. 2018, 8, 7270-7278).

Therefore, we conclude that the enhanced photocatalytic activity in 30%PEG@BT-COF compared to the BT-COF is not primarily caused by the Pt nanoparticles formed outside the 30%PEG@BT-COF.

The revision is shown in the main text, *Page 15, Line 5-14*, “The similar increase in the average size of Pt nanoparticles (2.2 nm) was observed when the short PEG chains ($M_w = 2000$) were threaded in the BT-COF (Supplementary Fig. 30). Thus, it is likely that more Pt nanoparticles are deposited on the surface of the PEG@BT-COF instead of in the pores. This feature should be beneficial for enhancing the photocatalytic performance. Nevertheless, the HER of 30%PEG-20k@BT-COF was much larger than those of the BT-COFs infiltrated with short PEG-2000 and PEG-400 chains (Supplementary Fig. 31). Therefore, the slight change of the deposition sites or sizes of Pt nanoparticles may not be the decisive factors responsible for the improvement in the photocatalytic performance of the 30%PEG-20k@BT-COF because the increasing sizes of Pt nanoparticles usually impair the photocatalytic activity⁴⁵.”

Supplementary Fig. 30 TEM images and statistical size distributions of the Pt nanoparticles photo-deposited on the BT-COF (a,b), 30%PEG-2000@BT-COF (c,d) and 30%PEG-20k@BT-COF (e,f), respectively.

Supplementary Fig. 31 Photocatalytic H₂ evolution rates of the BT-COF, PEG400@BT-COF, PEG2000@BT-COF and PEG20k@BT-COF, respectively, under $\lambda > 420$ nm irradiation for 8 h, using ascorbic acid as a sacrificial electron donor (0.1 M) in the presence of Pt nanoparticle (~3.7 wt.%). The feeding amounts of PEG400, PEG2000 and PEG20k are all 30wt% relative to the BT-COF.

Reviewer #3:

The authors have adequately answered all concerns of this reviewer and therefore I can recommend this work for publication.

As additional note: I'm slightly surprised by the relatively large amount of terminal groups in the COFs (see answer to comment #5). These cannot be quantified of course from CP-MAS NMR, but at least the peaks are clearly visible, which is not often seen for other COFs (or probably just not mentioned as in the first version of this work). The authors state that these peaks have been reported in an earlier work by themselves. I wonder if also other groups have seen pronounced peaks for terminal groups when using pyrrolidone or other catalysts?

Response: Thanks for pointing out this issue.

The use of pyrrolidine as a catalyst for the synthesis of crystalline 2D COFs has only been reported by our group (*e.g.* *Angew. Chem. Int. Ed.* **2016**, *55*, 13979; *Chem. Commun.* **2021**, *57*, 331). Thus, to the best of our knowledge, there are no related publications regarding the use of pyrrolidine as a catalyst for the synthesis of β -ketoenamine-linked COFs.

Currently, the most commonly used catalyst for the synthesis of COFs by forming the β -ketoenamine bonds is aqueous acetic acid solution (6M). As reported, the unreacted aldehydes

could also be found in the FT IR spectra, wherein a peak around 1695 cm^{-1} assigned to the -CHO group can be observed (e.g. *Nat. Commun.* **2020**, **11**, 497). Likewise, in the solid-state ^{13}C NMR spectra, the pronounced carbon signal ascribed to the unreacted aldehydes can be found in the range of 185~200 ppm (e.g. *Angew. Chem. Int. Ed.* **2020**, **59**, 16902; *Angew. Chem. Int. Ed.* **2021**, **60**, 5363).

Regarding the answer to comment #7. That the total water uptake capacity decreases after loading PEG into the pores of the COFs is not totally surprising. An increased hydrophilicity is however not seen necessarily by a change in total uptake but by the shape of the isotherm, especially water uptake at low pressures. Comparing the isotherms indeed a slightly higher uptake of water at lower pressures is seen for PEG@TP-COF, but probably this is not such pronounced that it must be further discussed or highlighted. I just think that the discussion of change in total water uptake is also not conducive when discussing the increased photocatalytic performance.

Response: Thanks for the insightful comment.

We agree with the reviewer that it is not necessary to correlate the total water uptake with the increased photocatalytic performances. As suggested by the reviewer, the change of hydrophilicity has been discussed on basis of the water uptake behaviors at low pressures. The corresponding revisions have been made in the main text, *Page 11, Line 5-9*, “As the mesopores of BT-COF and TP-COF were filled up with high-molecular-weight PEG chains, the total water uptake significantly decreased, whereas the isotherms at low pressures still remained at the similar level (Fig. 2e and Supplementary Fig. 23), indicative of the preservation of hydrophilicity originating from the infiltrated PEG chains.”

REVIEWERS' COMMENTS

Reviewer #2

I appreciate all the hard work and efforts by the authors for this revision. The authors could adequately address the concerns of this reviewer and now I would like to suggest this manuscript for publication in Nature Commun.